# Genomic analyses of 10,376 individuals in the Westlake BioBank for Chinese (WBBC) pilot project

Pei-Kuan Cong [1,2,3,11], Wei-Yang Bai[1,2,3,11], Jin-Chen Li [4,5,6,11], Meng-Yuan Yang[1,2,3], Saber Khederzadeh [1,2,3], Si-Rui Gai[1,2,3], Nan Li[7], Yu-Heng Liu[7], Shi-Hui Yu[8], Wei-Wei Zhao[8], Jun-Quan Liu[8], Yi Sun[8], Xiao-Wei Zhu[1,2,3], Pian-Pian Zhao[1,2,3], Jiang-Wei Xia[1,2,3], Peng-Lin Guan[1,2,3], Yu Qian[1,2,3], Jian-Guo Tao[1,2,3], Lin Xu[9], Geng Tian[9], Ping-Yu Wang[9], Shu-Yang Xie [9], Mo-Chang Qiu[10], Ke-Qi Liu[10], Bei-Sha Tang [4,5✉] & Hou-Feng Zheng [1,2,3✉]

We initiate the Westlake BioBank for Chinese (WBBC) pilot project with 4,535 whole-genome sequencing (WGS) individuals and 5,841 high-density genotyping individuals, and identify 81.5 million SNPs and INDELs, of which 38.5% are absent in dbSNP Build 151. We provide a population-specific reference panel and an online imputation server (https://wbbc.westlake.edu.cn/) which could yield substantial improvement of imputation performance in Chinese population, especially for low-frequency and rare variants. By analyzing the singleton density of the WGS data, we find selection signatures in *SNX29*, *DNAH1* and *WDR1* genes, and the derived alleles of the alcohol metabolism genes (*ADH1A* and *ADH1B*) emerge around 7,000 years ago and tend to be more common from 4,000 years ago in East Asia. Genetic evidence supports the corresponding geographical boundaries of the Qinling-Huaihe Line and Nanling Mountains, which separate the Han Chinese into subgroups, and we reveal that North Han was more homogeneous than South Han.

[1] Diseases & Population (DaP) Geninfo Lab, School of Life Sciences, Westlake University, Hangzhou, Zhejiang, China. [2] Westlake Laboratory of Life Sciences and Biomedicine, Hangzhou, Zhejiang, China. [3] Institute of Basic Medical Sciences, Westlake Institute for Advanced Study, Hangzhou, Zhejiang, China. [4] Department of Neurology, Xiangya Hospital, Central South University, Changsha, Hunan, China. [5] National Clinical Research Center for Geriatric Disorders, Department of Geriatrics, Xiangya Hospital, Central South University, Changsha, Hunan, China. [6] Center for Medical Genetics & Hunan Key Laboratory, School of Life Sciences, Central South University, Changsha, Hunan, China. [7] The High-Performance Computing Center, Westlake University, Hangzhou, Zhejiang, China. [8] Clinical Genome Center, KingMed Diagnostics, Co., Ltd., Guangzhou, Guangdong, China. [9] WBBC Shandong Center, Binzhou Medical University, Yantai, Shandong, China. [10] WBBC Jiangxi Center, Jiangxi Medical College, Shangrao, Jiangxi, China. [11] These authors contributed equally: Pei-Kuan Cong, Wei-Yang Bai, Jin-Chen Li. ✉email: bstang7398@163.com; zhenghoufeng@westlake.edu.cn

Understanding the architecture of the human genome has been a fundamental approach to precision medicine. Over the past decade, great progress has been made to unravel the genetic basis of complex traits/diseases[1] and the human evolutionary history[2]. The in-depth analysis of global populations with diverse ancestry could improve the understanding of the relationship between genomic variations and human diseases[3]. However, genetic studies exhibited a vast imbalance in the global population, with individuals of European descent took up ~79% of all genome-wide association study participants[3,4]. Similarly, most of the whole-genome sequencing (WGS) efforts were predominantly conducted on European populations, such as Dutch[5], UK[6], and Icelandic population[7]. Even in larger genomic projects such as the Trans-Omics for Precision Medicine (TOPMed) program, which consisted of ~155k participants from >80 different studies, only 9% of samples were of Asian descent[8,9]. Therefore, large-scale genomic data are required to understand the genetic basis of the Asian population. Recently, some studies have sequenced and analyzed the Asian populations including Japanese[10], Korean[11], and Chinese[12,13]. The Singapore SG10K pilot project reported 4810 whole-genome sequenced samples, including 903 Malays, 1127 Indians, and 2780 Chinese[14] and the pilot study of the GenomeAsia 100K Project presented a dataset of 1267 individuals from different countries across Asia[15].

Despite the above efforts, the Chinese population is still underrepresented in human genetic studies, which could increase the health disparities if Chinese personal genomes are underserved[16–18]. China, as the most populated country, is a multi-ethnic nation, in which the Han Chinese accounts for 90% of the population. Generally, the entire territory of the country includes 34 administrative divisions, including provinces, municipalities, and special administrative regions. Our previous study[19] demonstrated that, even with the Haplotype Reference Consortium reference panel which contained 64,976 human haplotypes[20], the imputation of the Chinese population could not reach the highest accuracy, a population-specific reference panel was still needed[19]. Therefore, the genetic study of the Chinese population has the potential to benefit ~20% of the world population, and provide a comparison to the rest of the world. Thus, we initiated the Westlake BioBank for Chinese (WBBC) project[21] to characterize the genomic variation and population structure in a large-scale cohort aiming to collect ~100,000 samples with deep phenotypes. Here, the genomic findings of the pilot project of the WBBC from 10,376 samples were described, covering 29 out of 34 administrative divisions of China.

## Results

**The WBBC pilot dataset and variants identified.** The WBBC pilot project sampled 10,376 individuals from 29 of 34 administrative divisions of the People's Republic of China (Fig. 1a and Supplementary Table 1). We performed WGS in 4535 individuals on NovaSeq 6000 platform. After removing contaminated and duplicated samples, 4480 individuals were retained for downstream analyses and statistics. The mean sequencing coverage was $13.9\times$, which covered 99.77% of the genome, with a range between $9.6\times$ and $65.2\times$ (Supplementary Fig. 1 and Supplementary Data 1). In addition, 6025 individuals were genotyped by high-density Infinium Asian Screening Array (ASA), including 184 individuals who were also whole-genome sequenced. We identified 81,498,995 variants (Ts/Tv = 2.15) after filtration from 103.96 million total raw variants, including 74,118,191 single-nucleotide variants (SNVs) and 7,380,804 insertions and deletions (INDELs). Of these variants, which the majority of variants (44.2 million, 54.5%) were singletons (Fig. 1b), 93.3% were rare (allele frequency, AF < 0.005) and low-frequency (AF = 0.005–0.05) variants (Supplementary Table 2). We provided a user-friendly website to search for the annotation and allele frequency of genetic variants in the Chinese population, including the four Han sub-regions (North, Central, South, and Lingnan). (https://wbbc.westlake.edu.cn/genotype.html).

Comparing the variants identified in WBBC with other existing databases, 45,696,726 variants were found not to present in the 1000 Genome Project (1KG)[22], gnomAD[23], or UK10K[6] (Fig. 1c). Of these, 45.6 (99.79%) million were rare variants (MAF < 0.005). We also found 31.37 (38.5%) million variants that were not present in dbSNP Build 151[24], including 29,015,419 SNVs and 2,353,726 INDELs. Of these variants, singletons accounted for 83.3%, and 99.97% of the variants (31.26 million) were rare with MAF < 0.005.

We assessed the SNV variants calling accuracy and sensitivity by comparison with SNP array data in the 184 individuals who were also whole-genome sequenced ($13.3\times–54.7\times$). Supposing the SNP array data as the true genotype, the heterozygote discordance rate of genotypes was reduced six-fold from 0.134 to 0.022 at $13.3\times$ sequencing depth and four-fold from 0.004 to 0.001 at $25.3\times$ after genotype refinement (Supplementary Fig. 2a). The non-reference (NR) genotype concordance rate extended to 99.88% at $25\times$ with increasing sequencing depth (Supplementary Fig. 2b). The NR sensitivity and specificity had an effective increase after genotype refinement with BEAGLE from 0.9211 to 0.9924 and from 0.9931 to 0.9999 (Supplementary Fig. 2c, d).

**Variants annotation and individual genome.** To characterize variants with a biological consequence, we annotated all the variants from 4480 individuals regardless of medical conditions with ANNOVAR tools[25]. For the variants not in dbSNP Build 151, the variants in intronic and intergenic regions took up 38.31% and 51.33% separately. Only 0.98% of variants are located in coding and splicing regions (Fig. 1d). The missense variants accounted for 54.22% of the variants in coding and splice regions, while splicing variants made up 29.69% of the variants (Fig. 1e). Among common variants, the number of synonymous variants was more than that of missense variants (Supplementary Table 2). However, the percentage of missense variants was distinctly higher than synonymous for the lower frequency and rare variants, consistent with the functional categories in the Northeast Asian Reference Database (NARD)[26] and Japanese individuals project (1KJPN)[10]. We also observed the markedly increasing of nonsense and frameshift variants among rare variants in accord with the finding in the Genome of the Netherlands project[5], which were signatures of population expansion and weak purifying selection (Supplementary Table 2). Interestingly, we also identified 1842 pathogenic or likely pathogenic variants recorded by ClinVar in our dataset regardless of medical conditions. Of these predicted disease-causing variants, 97.4% variants were rare, 1.7% variants were low frequency, and 0.9% were common variants, which arose from selection pressure subjected to these rare variants. The c.315-48T>C in *FECH* gene were common variants (MAF > 0.3) in the Chinese and Asian populations; however, the allele frequency was only 0.06 in the CEU population and gnomAD. This deep intronic variant was considered a pathogenic variant and might cause erythropoietic protoporphyria in the European patients[27].

We selected 1151 healthy individuals for the autosomal variants' statistic of a personal genome. On average, an individual carried 3,068,811 SNVs and 257,832 INDELs, including 9106 missense variants, 10 stop loss, 73 stop gain, and 190 frameshift or non-frameshift INDELs (Supplementary Table 3). The number

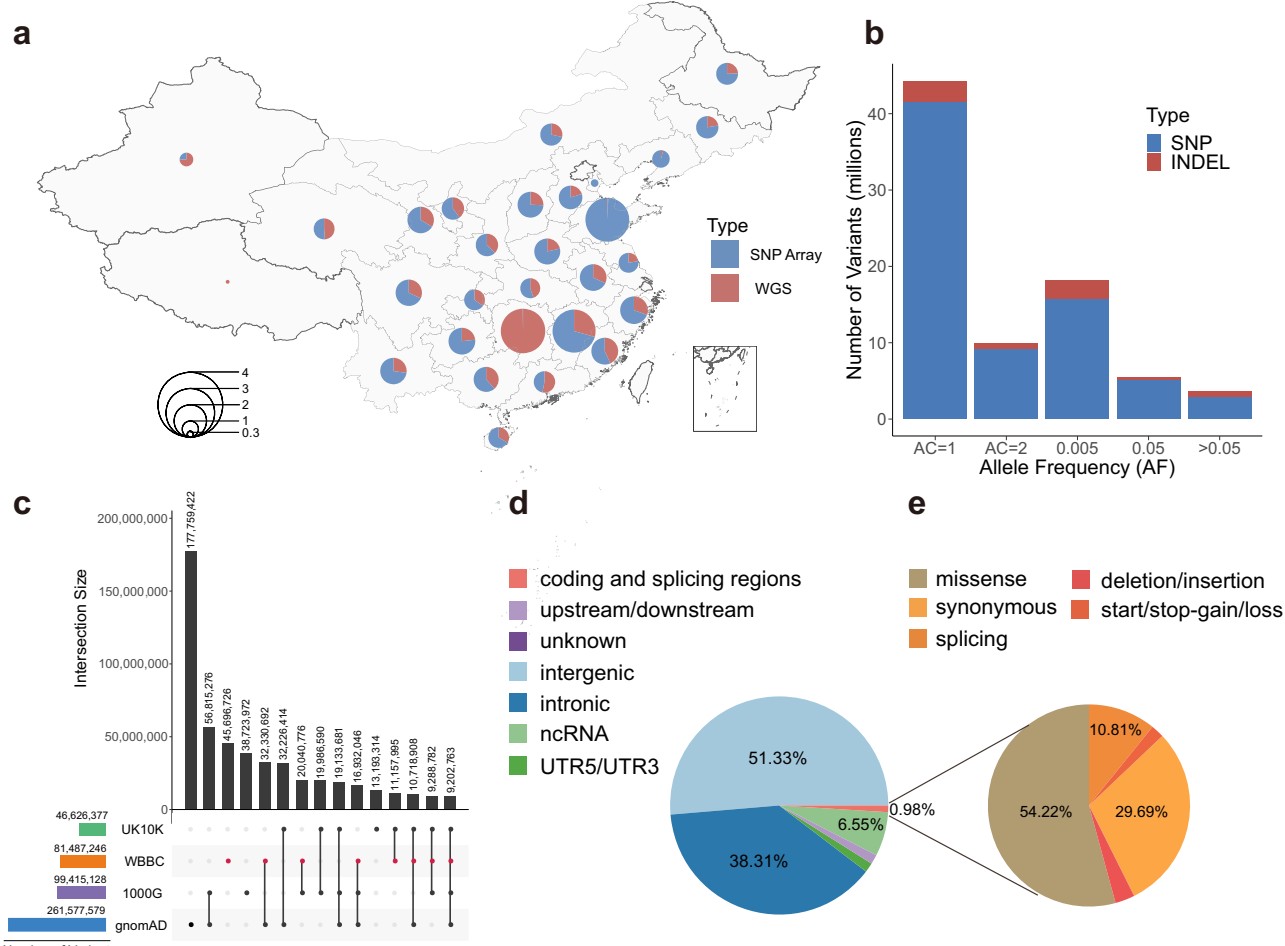

**Fig. 1 The statistics of samples and variants in the WBBC. a** Sample distribution and statistics by geography. The proportion of samples sequenced by whole-genome sequencing (WGS) and those genotyped by high-density Infinium Asian Screening Array (ASA) were marked in red and blue, respectively. **b** The number of SNV and INDEL variants identified in the WBBC cohort in five frequency bins: $AC = 1$, $AC = 2$, $AC > 2$ and $AF < 0.005$, $0.005 \leq AF \leq 0.05$, and $AF > 0.05$. **c** The number of variants in 22 autosomes and X chromosome in the WBBC, 1000 Genome Project (1000G), gnomAD, and UK10K datasets. The horizontal bar plot shows the total number of variants in each of the four datasets. The individual dots and connected dots indicate each dataset and a combination of two or more datasets, respectively. Each vertical bar represents the number of variants in each dataset or overlapping variants in those datasets. **d** Functional annotations of all variants that were absent in dbSNP Build 151. The proportion of each category was filled with a different color. **e** The pie chart only displayed the variants in the coding and splicing regions (10 bp from exon-intron boundary). Source data are provided as a Source Data file.

of variants in a Chinese genome was similar to the variants in the GoNL (~3 million)[5] and less than the number of SNVs in the UK10K project (3,222,597 variants)[6] and SG10K (3,308,882 variants)[14]. To measure the prevalence of pathogenic variants in a healthy individual, we annotated the variants by ClinVar[28]. In total, we identified 732 pathogenic or likely pathogenic variants in the healthy population, and on average each individual carried 11 pathogenic or likely pathogenic variants (het/hom = 2.08) (Supplementary Table 3 and Supplementary Data 2). Each genome carried $3.6 \pm 2.1$ (mean ± SD) pathogenic homozygote variants in the Chinese Han population, which was consistent with the value $(3.9 \pm 2.0)$ in Chinese in the SG10K pilot study and fewer than the number of pathogenic homozygote variants in Malays $(4.3 \pm 2.1)$ and Indians $(4.9 \pm 2.2)$[14]. The pathogenic variants p.V37I (*GJB2*, MAF = 0.098), p.K530X (*DUOX2*, MAF = 0.015), p.G1109Efs*13 (*FLG*, MAF = 0.013), and p.R266X (*SERPINB7*, MAF = 0.012) had a relatively high allele frequency in WBBC. However, these common pathogenic variants in the East Asian population were not observed in the European population in the 1000 Genome Project.

**Whole-genome-wide singleton density score analysis and selection inference**. The singleton density score (SDS) can be applied to infer recent allele frequency changes by calculating the distance between the nearest singletons on either side of a test-SNP using whole-genome sequence data[29]. We tried to infer the recent allele frequency changes at SNVs of the Han Chinese population by calculating SDS based on 4,258,941 bi-allelic SNVs and 17,951,337 singletons from 4334 whole-genome sequenced Han individuals. We found a significant selection signature in *SNX29* gene (Fig. 2a) on chromosome 16p, which encoded the sorting nexin-29 protein and was ubiquitously expressed in the kidney, lymph node, ovary, and thyroid gland tissues[30]. More than 30 SNPs on *SNX29* gene exhibited strong selection signatures ($p < 5 \times 10^{-8}$), which indicated significant enrichment of selection in this genomic region. Relatively higher derived allele frequency (DAF) was observed on the top SNP rs75431978 in the Han Chinese population (DAF = 0.181, $p = 5.54 \times 10^{-16}$), East Asian population (DAF = 0.146), Mongolian (DAF = 0.18), Korean (DAF = 0.187) and Japanese (DAF = 0.12), compared to the values obtained in 1000 Genome Project SAS (DAF = 0.062),

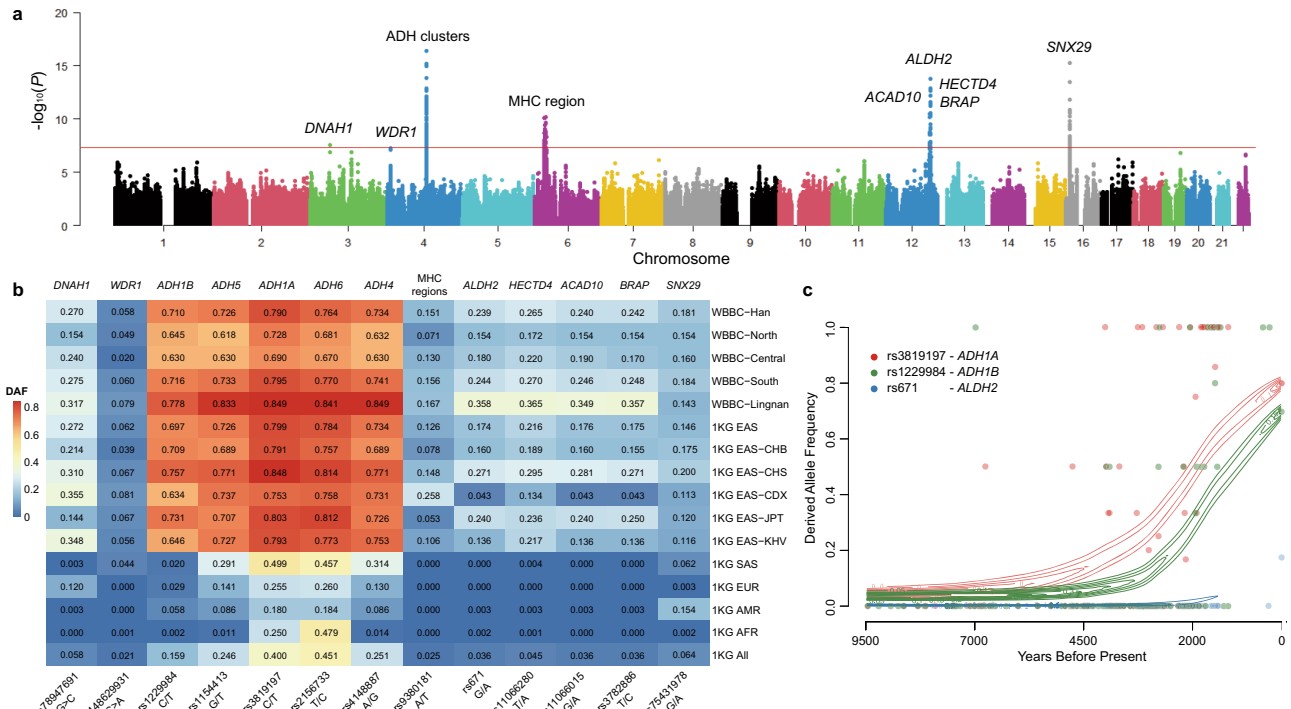

**Fig. 2 Whole-genome-wide recent selection signatures of the Han Chinese population by singleton density score (SDS) analysis. a** Manhattan plot of the natural selection signatures from the WGS data of the Han Chinese individuals. The y-axis represents the -log10 ($P$) of the two-tailed $p$ values for standardized SDS z-scores. The horizontal red line indicates the significance threshold ($p < 5 \times 10^{-8}$). **b** The derived allele frequency (DAF) of SNVs with significant selection signatures for different populations. The WBBC-Han is all the Han Chinese individuals sequenced by whole-genome sequencing (WGS) in the WBBC cohort. North, Central, South, and Lingnan are the four Han subgroups. EAS, SAS, EUR, AMR, and AFR come from the 1000 Genome Project (1KG). **c** The inferred allele frequency trajectory for the derived alleles at rs3819197, rs1229984, and rs671 over the past 9500 years from the ancient individuals of East Asia. The dot indicates the allele frequency in each generation (25 years/generation). Source data are provided as a Source Data file.

EUR (DAF = 0.003) and AFR (DAF = 0.002) populations (Fig. 2b). Although SNX29 was reported to be a biomarker for vasodilator-responsive pulmonary arterial hypertension[31] and major mental disorders[32], we found the selection signatures in SNX29 gene in both healthy and Parkinson's disease individuals. We also identified other two potential selection signals DNAH1 rs78947691 on chromosome 3 (DAF = 0.270, $p = 2.65 \times 10^{-8}$) and WDR1 rs148629931 (DAF = 0.058, $p = 5.44 \times 10^{-8}$) on chromosome 4 (Fig. 2a), and the $p$ values were very close to the significance threshold ($p < 5 \times 10^{-8}$). The top rs78947691 in the intron 16 of DNAH1 gene and rs148629931 in the upstream of WDR1 gene have relatively high DAF in the EAS population, comparing with other populations (Fig. 2b). The polymorphisms in the DNAH1 gene showed the potential association with male infertility in the Chinese[33,34], while the variation in the WDR1 gene was the risk factor for gout development in the Chinese population[35,36].

We also confirmed several significant natural selection signals at alcohol dehydrogenase (ADH) gene clusters (rs1229984, $p = 6.07 \times 10^{-16}$), the major histocompatibility complex (MHC) region (rs9380181, $p = 6.43 \times 10^{-11}$), and ALDH2 (rs671, $p = 1.68 \times 10^{-14}$) (Fig. 2a). These three selection signature regions have also been identified in the Japanese population[37]. The alcohol-metabolizing enzymes, such as the ADH genes (including ADH1A, ADH1B, ADH4, ADH5 and ADH6) and the aldehyde dehydrogenase (ALDH2) gene, had an effective impact on the alcohol metabolism pathway and the consequent alcoholism protective effect, which strongly indicated diverse ethnic-specific alcohol consumption patterns[38–42]. Similarly, the high DAF in these genomic loci, particularly rs1154413 (ADH5),

rs4148887 (ADH4), rs2156733 (ADH6), rs3819197 (ADH1A), rs1229984 (ADH1B), and rs671 (ALDH2) (0.726, 0.734, 0.764, 0.790, 0710 and 0.239, respectively), illustrated corresponding alleles associated with alcoholism in the Han Chinese, when compared with other non-East Asian (Fig. 2b and Supplementary Table 4). Interestingly, we observed a higher-level DAF in these SNVs in the South and Lingnan regions compared to the North and Central Han, which reflected the recent regional DAF changes and adaptation in this populous ethnicity and articulated different drinking habits or specific alcohol consumption. In addition, we identified other selection signals in chromosome 12, including rs11066280 ($p = 1.41 \times 10^{-13}$) in HECTD4 gene, rs11066015 ($p = 2.57 \times 10^{-12}$) in ACAD10 gene and rs3782886 ($p = 4.11 \times 10^{-12}$) in BRAP gene, which were limited to EAS ancestry populations. These three genes adjacent to the ALDH2 gene in chromosome 12q were within a large linkage disequilibrium (LD) block[43], revealing that the region had been under positive selection for a long time.

We estimated the allele frequency trajectories for ADH1A (rs3819197), ADH1B (rs1229984), and ALDH2 (rs671) with a hidden Markov model[44] using the allele frequency time-series data calculated from the East Asian ancient samples (9500–300 BP) and present-day individuals. The derived alleles of the SNP rs1229984 and rs3819197 emerged around 7000 years ago and tended to be more common from 4,000 years ago in the ancient population (Fig. 2c). The derived allele (A) of rs671 in ALDH2 gene was a common variant (MAF = 0.174) and strongly selected in the modern East Asian population, yet very rarely in non-East Asian (Fig. 2b); however, the allele A was absent in the East Asian ancient DNA data, suggesting a more recent selection.

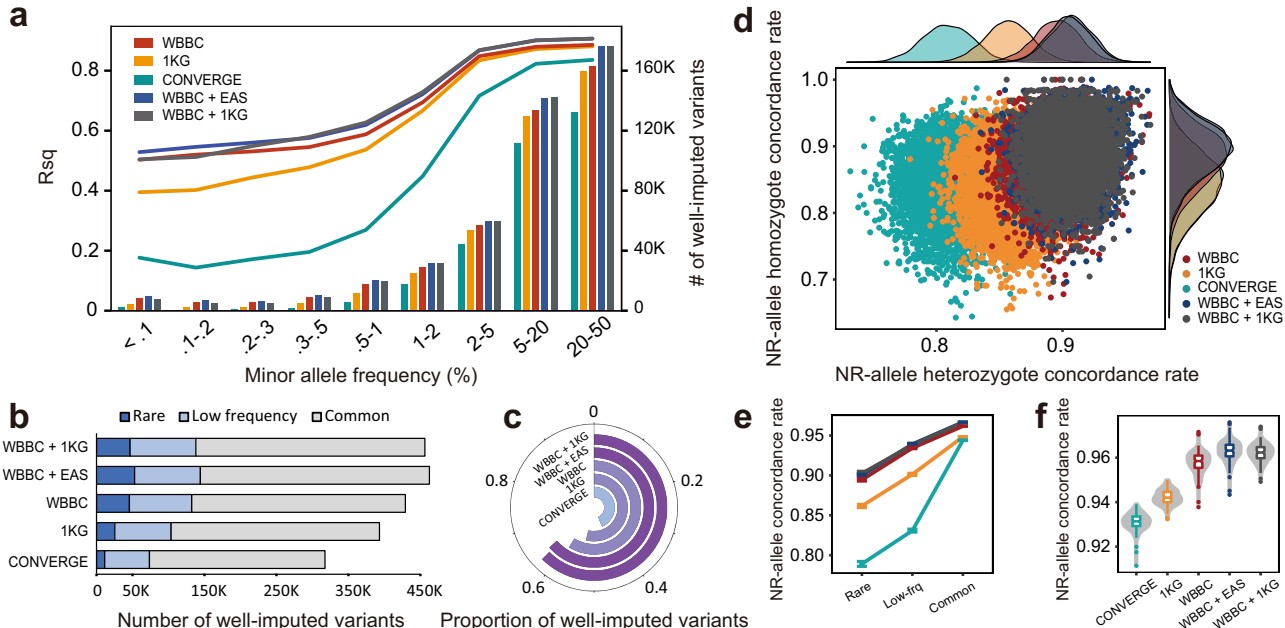

**Fig. 3 Imputation performance of five reference panels in the Han Chinese. a** The average $R$-square ($R$sq) and number of well-imputed ($R$sq ≥ 0.8) variants in shared sites of five reference panels (729,958 SNPs). All shared variants were grouped into nine MAF bins. **b**, **c** The cumulative number and proportion of well-imputed variants in shared sites of five panels, there were 729,958 shared SNPs in total. **d** Non-reference allele (NR-allele) concordance rate distribution (imputed variants vs. array variants). Each dot represents an individual. The plots on the top and right are the corresponding density distributions. **e**, **f** The NR-allele genotype concordance rate for rare, low-frequency, and common variants and overall variants (imputed variants vs. WGS variants). A total of 184 unrelated samples with both sequencing and genotyping data were used for the evaluation. The concordance rates for each variants group with mean value ± SEM, and quartile for each panel were plotted. The 1KG means 1000G Phase 3 and EAS means East Asian group in 1000G Phase 3. All imputations were conducted on chromosome 2. Source data are provided as a Source Data file.

**Imputation reference panel for the Chinese population**. We evaluated the genotype imputation accuracy of the WBBC, 1KG (Phase 3, v5a)[22], CONVERGE[45], and two combined reference panels (WBBC + EAS and WBBC + 1KG) in the Chinese population (Supplementary Fig. 3). The results showed that the WBBC panel, with almost fifteen-fold more Chinese samples than the 1KG Project, yielded substantial improvement for imputation for low-frequency and rare variants (Fig. 3a). The two combined panels, WBBC + EAS and WBBC + 1KG, almost tied and possessed both the highest $R$-square ($R$sq) and number of well-imputed variants in the shared sites with a MAF range of 0.2% to 50%, followed by the WBBC, 1KG, and CONVERGE (Fig. 3a). For the rare variants with MAF less than 0.2%, WBBC + EAS panel showed the best performance, and the WBBC panel performed roughly the same as the WBBC + 1KG (Fig. 3a). This result indicated that merging EAS individuals of the 1KG to increase the haplotype size of the WBBC could improve panel's performance across all MAF bins, but merging the whole 1KG cannot yield more improvement than merging-EAS-only and even not equal to it when the imputed variants were quite rare. Taking all shared variants together, the WBBC + EAS yielded the most well-imputed variants in shared sites, while the CONVERGE panel imputed the least (Fig. 3b). The proportion of imputed variants with $R$sq ≥ 0.8 for CONVERGE was the only one under 50% across five panels, even it was population-specific to Chinese (Fig. 3c), indicative of the importance of coverage sequencing depth of a reference panel.

To comprehensively evaluate the imputation accuracy for the five panels, we further calculated the NR genotype concordance rate between imputed and genotyped variants by chip array and WGS respectively (imputation vs. chip array and imputation vs. WGS). Two combined panels had the most promising distributions of the NR concordance rates, which were almost coincident with

each other, indicating that the NR concordance rates for Chinese imputation could barely benefit from the extra non-East Asian haplotypes of the reference panel (Fig. 3d). Besides, we could know that the peaks of two combined panels in density plots were higher than other panels, indicating that the distributions of NR concordance rates were more concentrated in the two combined panels (Fig. 3d). The performance of the WBBC panel was slightly behind the two combined panels, but was superior to the 1KG and CONVERGE (Fig. 3d). We also calculated the NR-allele concordance rate between the imputed genotypes and the directly sequenced genotypes. Not surprisingly, the two combined panels performed best and were approximately coincident and very closely followed by the WBBC (Fig. 3e). This result suggested that the improvement provided by the EAS and 1KG were unremarkable. Considering all variants together, the WBBC + EAS panel showed the highest NR-allele concordance rate, followed by the WBBC + 1KG, WBBC, 1KG and CONVERGE (Fig. 3f).

Overall, we employed $R$sq, and NR-allele concordance rate for both WGS and array genotype to measure the imputation accuracy for the five panels. Our results demonstrated the superiority of the WBBC as a reference panel for Chinese population imputation. Compared to the 1KG and CONVERGE, WBBC panel greatly improved the imputation accuracy, especially for the rare and low-frequency variants. Besides, merging EAS/1KG haplotypes into the WBBC could further improve the imputation accuracy.

To facilitate genotype imputation in the Chinese population, we developed an imputation server with user-friendly website interface for public use (https://imputationserver.westlake.edu.cn/). Users can register and create imputation jobs freely by uploading their bgzipped array data (VCF-formatted) to our server under a strict policy of data security. To ensure the integrity of array data for the next phasing and imputation, some basic QC should be performed,

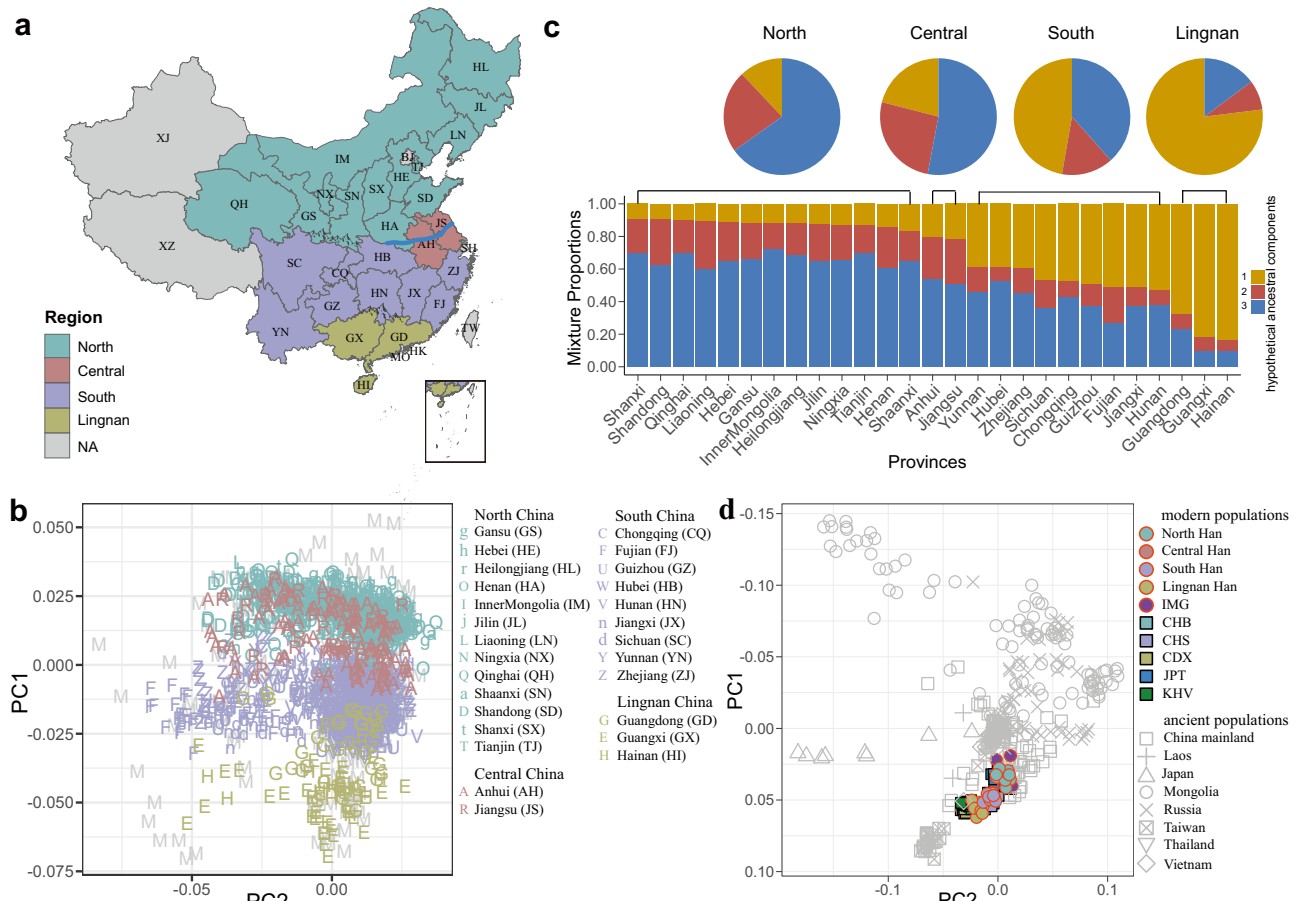

**Fig. 4 PCA and ADMIXTURE analysis of the Han Chinese populations and East Asians. a** A map of the People's Republic of China showing its 34 administrative divisions. "NA" indicates that the Han Chinese samples were not recruited from that region. The Qinling-Huaihe River line lies in central China, while the Nanling Mountains are in southern China. **b** Principal component analysis (PCA) of the Han and Minority Chinese individuals from four sub-regions. The administrative divisions are shown by the distinct letters. Minority individuals are marked with "M". The Han Chinese populations can be classified into four subgroups: North Han (cyan color), Central Han (dark-red color), South Han (purple color), and Lingnan Han (golden color). **c** ADMIXTURE analysis of 2056 Han Chinese individuals from 27 administrative divisions for the optimal K value = 3. Each vertical bar represents the average proportion of ancestral components in the regions. The length of each color indicates the percentage of inferred ancestry components from ancestral populations. The upper pie charts denote the average proportion of components across individuals from the four subgroups. **d** Plots of the first two principal components for modern and ancient East Asian individuals. Source data are provided as a Source Data file.

such as removing mismatched SNPs, monomorphism, and duplicate SNPs. The server provided a choice of four reference panels to conduct the imputation, including the WBBC, 1KG Phase 3, WBBC combined with EAS, and WBBC combined 1KG Phase 3. All panels in both GRCh37 and GRCh38 were built to meet different needs. Besides, service of phasing was also provided in our server for users who cannot afford the corresponding heavy computational load. An email reminder will be sent to the user when the imputation job is finished, and then user can download the imputed genotype data and the corresponding statistics file with an encrypted link. The SHAPEIT and MINIMAC were employed in our server for phasing and imputation, respectively. More details including the policy of data security, statistics of four reference panels, and the reference manual were specified in our website.

**Genetic evidence supported the geographical boundaries of the Qinling-Huaihe Line and Nanling Mountains.** To explore the Chinese population structure, we performed principal component analysis (PCA) on 2056 Han Chinese individuals and 205 minority individuals from 29 of 34 administrative divisions of China (Fig. 4a). PC1 and PC2 revealed the main genetic structure of the

Chinese population, with PC1 displaying a population stratification along the north-south cline, reflecting the geographical locations (Fig. 4b). The genetic difference of the Han population corresponded to the geographical boundaries of the Qinling-Huaihe River Line and Nanling Mountains. Based on the PCA analysis and traditional geographical boundaries in China, the Han Chinese could be classified into four subgroups: the North Han (Gansu, Hebei, Heilongjiang, Henan, Inner Mongolia, Jilin, Liaoning, Ningxia, Qinghai, Shaanxi, Shandong, Shanxi, and Tianjin) (Supplementary Figs. 4a and 5), the Central Han (Anhui and Jiangsu) (Supplementary Figs. 4b and 6), the South Han (Chongqing, Fujian, Guizhou, Hubei, Hunan, Jiangxi, Sichuan, Yunnan, and Zhejiang) (Supplementary Figs. 4c and 7), and the Lingnan Han (Guangxi, Guangdong and Hainan) (Supplementary Figs. 4d and 8). The average pairwise $F_{ST}$ values within subgroups were 0.00024, 0.00023, 0.00037, and 0.00059 for North Han, Central Han, South Han, and Lingnan Han, respectively (Supplementary Table 5). The pairwise $F_{ST}$ between Central and North Han (0.00021) was much lower than the $F_{ST}$ between Central and South Han (0.00083), between Central and Lingnan Han (0.00386) (Supplementary Table 6), indicating that Central Han was overlapping with North and South Han, and more close

to North Han. North-South Han (0.0015) and South-Lingnan Han (0.0013) had relatively high $F_{ST}$ values and were classified into different subgroups. PC3 and PC4 displayed no discernable geographical structure and subpopulations (Supplementary Fig. 4e). When the 104 JPT (Japanese in Tokyo, Japan) and 99 KHV (Kinh in Ho Chi Minh City, Vietnam) samples were included, the KHV population formed a cluster overlapping with Lingnan Han, while the JPT population was closer to the North Han Chinese (Supplementary Fig. 4f).

We estimated the ancestral composition of the Han Chinese population from 27 provinces using the ADMIXTURE program (Fig. 4c). The average number of presumed ancestral populations was calculated in each province with the optimal $K = 3$. The ancestry proportion of the North Han accounted for about 66% on component 3. The ancestral component of the Central Han was closer to the North Han with 52.1% on component 3, while the admixture components in the South Han were 46.3% on component 1 and 40% on component 3, respectively, which did not show the predominant ancestral components. We found a distinctly higher proportion of component 1 in Lingnan Han, at 78% of ancestry composition compared to other ancestral components. North Han, South Han, and Lingnan Han showed significantly different clusters, while central Han embodied the ancestral components of both northern and southern populations. We combined the East Asian data from 1KG and conducted ADMIXTURE analysis from $K = 2$ to $K = 8$ (Supplementary Fig. 9). Most components of the CHB population were consistent with North Han, except for part of samples originating from South Han, while the CHS population mainly came from South Han. From $K = 2$ to $K = 6$, the JPT population and Han Chinese population were classified into different groups, but more close to the North Han at $K = 2$. KHV population clustered with Lingnan Han at $K = 2$, whereas from $K = 3$ onwards, the KHV population and Lingnan Han clustered separately.

We collected 396 published ancient genomes from 8 countries or regions from 40,000 to 300 years ago and 95 representative present-day genomes to reveal the population relationships between modern and ancient individuals in East Asia (Fig. 4d). We calculated the pairwise $F_{ST}$ values to evaluate genetic differentiation within modern and ancient populations (Supplementary Table 7). The ancient population exhibited the larger genetic distances from 0.00365 to 0.25656 with a median value of 0.03242, while the modern population had the values from 0 to 0.01885 (Supplementary Fig. 10a). The genetic distance between northern and southern subgroups in modern Chinese population (0.00195) distinctly lower than the distance of 0.01902 in ancient Chinese population. Genetic divergence increased with geographic distance and was more noticeable in the ancient than modern populations (Supplementary Fig. 10a). The PCA analysis showed that there were strong genetic differences between ancient individuals in the northern and southern areas. The ancient individuals from North Asia (e.g., Mongolia and Russia) were closer to modern North Han than South Han, while both modern and ancient samples from the Southern area (South, Lingnan, Taiwan, Thailand, and Vietnam) were closely clustered together, which was consistently fit well with the geographic distribution of the populations. The 144 ancient individuals from the China mainland were mostly close to the modern North Han, and there was population stratification with the modern Chinese population in the PCA analysis, which suggested the human migration and admixture in Northern and Southern China during the long population history of East Asia[46,47].

**Population structure and demographic history in four sub-regions of the Han Chinese population**. Weir–Cockerham $F_{ST}$ is an allele frequency-based metric to measure the population differentiation due to genetic structure. We calculated pairwise $F_{ST}$ and performed hierarchical clustering for 27 administrative divisions of China and 26 populations of the 1KG. The 27 administrative divisions were mainly clustered into three groups and showed an association with geography (Fig. 5a and Supplementary Table 5). Anhui and Jiangsu provinces, which we designated as the Central region of China, were clustered with Northern provinces, indicative of a closer genetic relationship. The other two groups, South and Lingnan, aligned with the regions we designated. Besides, the hierarchical branches suggested that the population differentiation between South and North was smaller than that between the South and Lingnan (Fig. 5a), reflecting the relatively shorter genetic distance. The two most remote regions in geography, North and Lingnan, were also found to have the largest population differentiation (Fig. 5a). Not surprisingly, the pairwise $F_{ST}$ clustering results between the WBBC and 1KG populations showed that the four designated regions were clustered into the East Asian (EAS) group (Supplementary Fig. 10b and Supplementary Table 6). In particular, North and Central were clustered with CHB, while South and Lingnan were clustered with CHS (Supplementary Fig. 10b). Using the four 1KG continent-level ancestry groups (AFR, EUR, AMR, and SAS) as the Non-Chinese population reference, we further investigated the geographical patterns of $F_{ST}$ in 27 administrative divisions of China. The AFR group showed the largest $F_{ST}$ that ranged from 0.14 to 0.15 (Supplementary Fig. 11b and Supplementary Table 8), indicative of the greatest population differentiation to the WBBC, while the SAS and AMR group yielded the least value (Supplementary Fig. 11a, c). The geographical patterns of $F_{ST}$ across the four sub-regions were similar to each other. On average, the Han Chinese in the Northern provinces had the relatively closer genetic structure to the Non-Chinese populations of the 1KG. Interestingly, Qinghai province was conspicuously highlighted in the geographic heatmaps, as its pairwise $F_{ST}$ value was obviously smaller than that of other administrative divisions (Supplementary Fig. 11), indicative of the genetic structure particularity of the Qinghai Han Chinese. Qinghai province is actually located in the high altitude region of China (average altitude >3000 m), and the genetic structure of Qinghai Han Chinese might be adapted to the local environment.

Next, we detected the identity-by-descent (IBD) segments with the logarithm of the odds (LOD) score >3 across individuals in the WBBC[48]. Unlike the Weir–Cockerham $F_{ST}$, IBD analysis is a haplotype-based approach to reveal the genetic structure and investigate the common ancestry of populations. The total IBD segment counts in each pair of administrative divisions were normalized by the corresponding sample size. We then performed the hierarchical clustering based on the matrix of normalized pairwise IBD counts. Similar to the results of $F_{ST}$ clustering, 27 administrative divisions were also mainly clustered into three groups, and individuals from Anhui and Jiangsu provinces were clustered in North (Fig. 5a, b). Besides, the results showed that most Southern provinces shared more IBD segments with Northern provinces than with Lingnan (Fig. 5b), just as observed in the Fst analysis (Fig. 5a), suggesting that the Han Chinese in South and North shared more common ancestry than South and Lingnan. The Fujian and Hunan, which we designated as the Southern province, had been found that joined up with Lingnan provinces by the multiple hierarchical branches, indicating that they were more close to Lingnan in ancestry (Fig. 5b), and contiguous to Lingnan geographically (Fig. 5c).

We inferred the history of effective population size for the Han Chinese, and the results across the four regions were shown in Fig. 5d. In the period from 1 million years ago to ~6 thousand years ago (kya), the Han Chinese size histories of four regions experienced almost identical dynamics. From 200 to ~10 kya, the

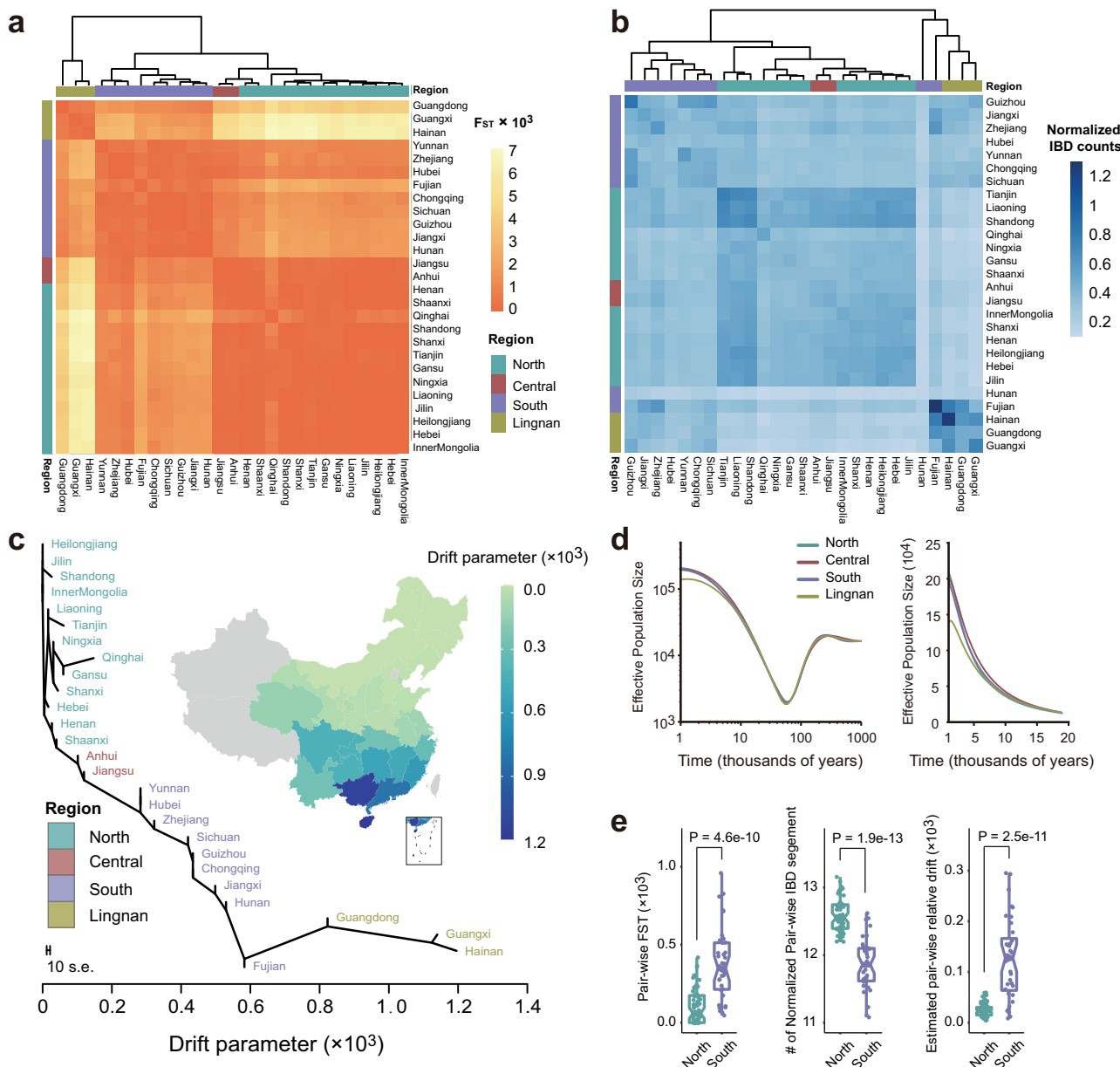

**Fig. 5 $F_{ST}$, IBD, genetic drift, and effective population size of the Han Chinese populations. a** A heatmap of pairwise $F_{ST}$ between any two of the 27 administrative divisions in China. The bars on the top and left show the classification of administrative divisions in the four regions. **b** A heatmap of pairwise IBD segments count between administrative divisions in China. The number of IBD segments is normalized by the sample size of each province. **c** A maximum-likelihood tree of the Han Chinese in 27 administrative divisions. The plot is rooted in the northernmost province, and the x-axis represents estimated genetic drift. All administrative divisions in the tree are colored by different regions. **d** Dynamics of effective population sizes of the Han Chinese in four regions. The x-axis means the thousands of years before present. The left panel shows the results on a log-log scale from 1 million to 1000 years ago and the right panel shows the results on a linear scale over the past 20,000 years. **e** Wilcoxon rank-sum test (two-sided) results for the $F_{ST}$ (left panel), normalized IBD segments (middle panel), and relative genetic drift (right panel) between pairwise Northern provinces and pairwise Southern provinces. The quartile for corresponding differences between pairwise provinces was plotted. A total of 12 Northern and 9 Southern provinces were included here. Source data are provided as a Source Data file.

effective population size experienced a steep decline and then grew rapidly, with the lowest point reached at ~60 kya, which was indicative of a bottleneck, consistent with previous demographic history studies[14,22,49]. Around 6 kya, the size histories of the Han Chinese from the Lingnan began to deviate from the other three regions, potentially reflecting the existence of a population substructure within the Lingnan Han Chinese (Fig. 5d).

Using the Han Chinese in the most northern province (Heilongjiang) of China as the reference, we estimated relative genetic drifts and inferred a rooted maximum-likelihood tree

between 27 administrative divisions by TreeMix software[50]. In the result shown in Fig. 5c, the relative drift of the provinces and municipalities were in line with the geographic location. To gain a better understanding of the result, we further drew a geographic heatmap that suggested a general genetic drift trend from the North to Lingnan, with the drift parameter increasing as the latitude decreased (Fig. 5c). To judge the confidence in the trend and tree topology, we performed ten bootstrap replicates by resampling blocks of SNPs. The trend was repeated in all replicate results (Supplementary Fig. 12). Besides, we found that the tree

topology of administrative divisions in Central, South, and Lingnan was stable. In the North, however, the tree topology was slightly different across the replicates, indicating that the genetic structures of the Northern administrative divisions were very similar and could not be precisely presented in the tree topology (Supplementary Fig. 12).

Enlightened by the genetic drift estimation results, we further investigated the homogeneity degree in the genetic structure of the Northern and Southern Han Chinese respectively. We performed the Wilcoxon rank-sum tests[51] for Northern and Southern administrative divisions using their respective pairwise $F_{ST}$ values, normalized IBD segments counts and relative drift parameters. The results showed that the Han Chinese from North had smaller population differentiation ($p = 4.6E–10$) and genetic drifts ($p = 2.5E–11$), and shared more IBD segments with each other ($p = 1.9E–13$) than those from South (Fig. 5e). These results suggested that the genetic structure of the Han Chinese in North was significantly homogeneous than those in South.

**Signatures of recent positive selection in four sub-regions of the Han Chinese population**. We employed the integrated haplotype score (iHS) test to identify recent natural signatures of positive selective sweeps in the North, Central South, and Lingnan Han populations[52]. The top 1% genomic regions with higher |iHS| scores were found in each population (Supplementary Data 3–6). The numbers of overlapping genomic windows of selective sweep regions across the four populations were shown in Supplementary Fig. 13. Only 34 (26%) sweep regions were found in all four populations. Most regions were shared in two or three of the four subgroups. Averagely, 23.2% of the regions were independent in North, Central, and South Han. However, the Lingnan Han had distinctly excess independent sweeps (50, 38.5%), which might be inherited from separate ancestral components, consistent with the conclusion from our demographic history analysis. Importantly, we observed the EDAR gene in the first three sweep regions in all four subgroups, which is a genetic determinant of hair thickness and has been under the strong selection pressure in East Asian[53–55]. A large genomic region extending for at least 285 kb on chromosome 7 in the top 10 window regions in all four subgroups contained eight contiguous robust selection genes (EPHB6, KEL, LLCFC1, MTRNR2L6, PRSS1, PRSS3P2, TRPV5, and TRPV6; Supplementary Data 3–6). The signature of positive selection of this region has been described only in European-Americans and not in the African-American population[56]. Our finding indicates that the interesting candidate region is not population specific.

We conducted Gene Ontology (GO) and Kyoto Encyclopedia of Genes and Genomes (KEGG) pathway analysis for candidate genes in the top 1% genomic regions with signals of recent selection by iHS (Supplementary Tables 9 and 10). The terms were selected according to p value (<0.05). The results of the GO analysis showed a significant enrichment of positively selected genes for ethanol metabolic process and ethanol oxidation in four sub-regions (Supplementary Table 9), consistent with the selective signatures by whole-genome-wide SDS analysis in the Han Chinese population. We also observed intriguing enrichment of keratinocyte differentiation, epidermal cell differentiation and skin development (CTSL, COL7A1, PKP3, SEC24B, SLITRK6, WNT10A, KRT10, KRT12, KRT20, and KRT23) in the South and Lingnan Han, which were not present in the North and Central Han populations. The KEGG analysis found that 23 pathways were enriched in the Han population with adjusted p values <0.05 (Supplementary Table 10). Of these pathways, Southern individuals displayed significantly enriched terms more than the northern

population. Tyrosine metabolism, retinol metabolism and fatty acid degradation were identified in four sub-regions.

## Discussion

We initiated the WBBC pilot project and performed the WGS of 4535 individuals from 29 of 34 administrative divisions of China. We provided a comprehensive map of the genomic variations for the Chinese population (https://wbbc.westlake.edu.cn/genotype.html). In addition, we found that the genetic evidence supported the geographical boundaries of the Qinling-Huaihe Line and Nanling Mountains, which separated the Chinese into four sub-regions (North, Central, South and Lingnan). The genetic architecture within North Han was more homogeneous than South Han, and the history of effective population size of Lingnan began to deviate from the other three regions about 6000 years ago. Furthermore, we found significant selection signatures around DNAH1, WDR1, and SNX29 genes in the Han Chinese, and confirmed selection signals at alcohol metabolism genes, and the selection of ADH1A and ADH1B strengthened from about 4000 years ago. We observed enriched positive selective sweeps of keratinocyte differentiation, epidermal cell differentiation, and skin development in the South and Lingnan Han. Finally, we provided a comprehensive reference panel for genotype imputation for the Chinese and Asian populations, and an online imputation server (https://imputationserver.westlake.edu.cn/) is publicly available now for genotype imputation.

In our study, we identified several significant selection signatures in DNAH1, WDR1, and SNX29 genes in the Han Chinese and confirmed several positive selection signatures using the singleton density-based and haplotype-based methods. The ethanol oxidation was the most significant enrichment in the Han Chinese population, which included the ADH gene cluster and ALDH gene cluster. The positive selection signatures on ADH1A and ADH1B were confirmed in the East Asian population[52,57] and in the Japanese population[37]. No significant selection signal in the ADH-cluster gene was detected in the population in Ulaanbaatar[58] and Tibetans[59]. In our study, the derived alleles of ADH1A and ADH1B gene emerged around 7000 years ago and tended to be more common from 4000 years ago in the ancient population, which coincided with the period from the late Neolithic to Bronze Age in China. The alleles experienced strong selection and might be driven by the advances in agricultural and wine production technology, which resulted in the prevalence of derived alleles. Interestingly, Peng et al. estimated the allele age in the East Asian population and reported that the emergence of the ADH1B (rs1229984) occurred about 10,000~7000 years ago[60], and another study suggested that the selection in the ADH1B gene intensified around 4000 years ago in the northern East Asia[61].

The MHC genes carried the significant selection signatures for adaptive autoimmune and infectious diseases in all populations shaped by natural selection over a long time[62]. The ectodysplasin A1 receptor (EDAR) encodes a member of the tumor necrosis factor receptor family, which was involved in the development of hair, teeth, and glands[63–65]. The SNV rs3827760 (NM_022336: c.1109T>C p.Val370Ala) in EDAR gene had been under strong natural selection in WBBC with a very high allele frequency (0.93). This non-synonymous SNV displayed a higher |iHS| score in four groups (North: 3.08, Central: 3.15, South: 3.15, and Lingnan: 3.65), which indicated the strongest signal of positive selection in this genomic region. Hunan and Jiangxi provinces are contiguous in the southern region of China, which took up 87% of all WGS samples and the positive selection signals might be diverse in other regions. We have conducted the subgroup analyses with a small number of samples (less than 800 individuals).

Due to the insufficient samples, only the *SNX29*, MHC region, ADH cluster, and *ALDH* gene cluster were detected. The SDS analyses might be influenced by the sample size. In addition, *SNX29* gene was reported to be a susceptibility gene associated with schizophrenia, bipolar disorder, or major depressive disorder disorders[32]. However, the significant selection signatures of *SNX29* gene were observed within healthy individual's random subgroups in our study. The biological mechanisms of the *SNX29* gene remain unclear.

The genetic structure of a population defines the level and extent of genetic variation within its constituent subpopulations. Although the genetic structure of north-south differentiation in the Chinese population was consistently observed in previous studies[12,13,66–68], subgrouping of the administrative divisions was not always consistent. For example, Hubei province was grouped into central in Cao et al.[12], while it was clustered into south in Xu et al.[66]. Our finding demonstrated that the Han Chinese populations were divided into four sub-regions (North, Central, South and Lingnan), which corresponds to the geographical boundary, the Qinling-Huaihe Line, and Nanling Mountains (Five Ridges). The Qinling Mountains are the east-west mountain range that stretches across the south of Gansu and Shaanxi provinces. The ~1000-km-long Huaihe River flows through the south of Henan province and the middle of Anhui and Jiangsu provinces. To some extent, the climate, culture, lifestyle, and cuisine between the Northern and Southern regions were different. Lingnan area is the region in the south of Nanling mountains (with five ridges) and the southeast of Yunnan-Guizhou Plateau in southern China, which refers to the administrative divisions of Guandong, Guangxi, Hainan, Hong Kong, and Macao[69]. Xu et al. showed that the Han Chinese was distinguished with three clusters corresponding roughly to northern Han, central Han, and southern Han[66]. Notably, the administrative divisions of North Han and Central Han by Xu et al. were consistent with our results; however, the southern Han would be accurately separated into South Han and Lingnan Han by the geographical barrier of the Nanling Mountains and Yunnan-Guizhou Plateau, which had been confirmed by our PCA and ADMIXTURE results. In addition, the genetic architecture within North Han was distinctly homogeneous, while the ancestral components of admixture in South Han were more diverse. Due to the absence of Han samples in seven administrative divisions (Beijing, Shanghai, Tibet, Xinjiang, Taiwan, Hong Kong and Macao), we have not inferred the population structure in these areas. The Qinling-Huaihe line is a boundary between semi-humid warm temperate continental monsoon climate and humid subtropical monsoon climate in China. The enrichment differences of candidate genes on skin development-related traits between northern and southern Han Chinese population might be the results of adaptive pressures selection, including the effects of geography, climate, and human migration. The epidermis is the outermost layer of the skin, which protects the body against pathogens and ultraviolet radiation, and is under adaptive pressure from sunlight duration and intensity.

Finally, using *Rsq* and NR-allele concordance rate metrics, we evaluated and compared the genotype imputation performance of the WBBC pilot with two existing panels, the 1KG Phase 3 and CONVERGE. Besides, given that the haplotype size of a panel and the genetic background between the panel and array are two crucial factors for imputation accuracy[19,70], we built and evaluated two more combined panels that merged the WBBC with the 1KG and EAS group by the reciprocal imputation approach[71]. The 1KG Project, which consisted of 2504 individuals from 26 worldwide populations, is the most diverse and commonly used panel for genotype imputation due to its high quality[22]. The CONVERGE is a population-specific reference panel for Chinese imputation but with low-coverage sequencing depth[45]. In our

study, the WBBC panel yielded a substantial improvement in imputation accuracy for low-frequency and rare variants than these two existing panels. Comparing to the CONVERGE panel, we had less samples in the WBBC; however, the higher coverage sequencing depth improved the genotype accuracy for variant calling. The WBBC + EAS and WBBC + 1KG panels performed better than WBBC panel alone, and the WBBC + EAS panel yielded the highest imputation accuracy for rare variants, the most well-imputed variants and the highest proportion of well-imputed variants. This observation was consistent with and further expanded our previous finding that population-specificity between reference panel and the imputed array was reasonably rigorous for the Han Chinese genotype imputation, and the accuracy benefited from the increasing of haplotype size via extra diverse individuals was limited, especially for rare variants[19]. Here, to maximize utilization of the WBBC pilot, we provided a large population-specific Genotype Imputation Server, which included the WBBC, 1KG, and the two combined reference panels for Chinese sample imputation.

In summary, we characterized large-scale genomic variations in the Chinese population and provided comprehensive genetic evidence for the geographical boundaries of the Qinling-Huaihe line and Nanling Mountains to divide the Han Chinese population into four subgroups, which could be helpful to the case–control design of the association study in the Chinese population. We elucidated the regional genetic structure and signatures of recent positive selection differences in modern and ancient individuals in East Asia. We also created a user-friendly website and high-performance genotype imputation server for East Asian samples. The online resource would practically be important for the genomic variants filtration of monogenic diseases and consequent association with complex traits in the population genetics field.

## Methods

**Study samples.** Both Westlake University and Xiangya Hospital contributed to the sample collection. The WBBC pilot project[21] of Westlake University has enrolled 14,726 individuals (4751 males and 9975 females aged 14–25 years, Supplementary Data 1) with diverse traits across 29 of 34 administrative divisions in China (provinces, municipalities, and special administrative regions), following the regulations of the Human Genetic Resources Administration of China (HGRAC). The Xiangya Hospital contributed another 3335 individuals (1653 males and 1682 females aged 51–89 years, Supplementary Data 1) including 1973 patients with Parkinson's disease and 1362 health controls; these samples were included in WBBC later on. Additional participant compensation was not provided. Of these samples, a total of 4535 individuals were whole-genome sequenced using NovaSeq 6000 Sequencing System (Illumina, Inc.), and 6025 individuals (including 184 WGS samples) were genotyped by high-density Infinium ASA (Illumina, Inc.) with 750 K variants (Supplementary Table 1). Hunan and Jiangxi provinces accounted for 87% of all WGS samples. Shandong ($n = 2801$) and Jiangxi ($n = 1730$) provinces comprised 77.6% of all ASA genotyped samples. All the participants signed the informed consent. The research program was approved by the HGRAC (2019-1962 and 2021-CJ1139), the Institutional Review Board of the Westlake University (2018-006), and Xiangya Hospital of Central South University (202005124).

**Whole-genome sequencing and variants calling.** Genomic DNA was extracted from peripheral blood samples collected from all the participants using the blood DNA extraction kit (TianGen Biotech, China). Library preparation was performed by NEXTflex Rapid DNA-Seq Kit (Bioo Scientific) following the standard protocol. WGS was conducted on the Illumina NovaSeq 6000 system at the KingMed Diagnostics Co. Ltd. The target depth was ~13× per individual, with about 40 GB sequencing data. Variants calling were conducted on all the samples via BWA version 0.7.17[72] and GATK4 version 4.1.4.0 (Supplementary Fig. 14)[73].

We performed the variants calling following the Genome Analysis Toolkit (GATK) best practices pipeline[73]. The reference genome GRCh38 were downloaded from the resource bundle on the GATK website (ftp://gsapubftp-anonymous@ftp.broadinstitute.org/bundle/). The reads in each lane were aligned to the human reference genome via the BWA mem tool to produce the SAM files. We used SAMtools (v1.7) view tool to convert SAM format files into BAM files[74] and GATK4 MergeSamFiles tool to sort and merge multiple lanes data into one bam. The MarkDuplicates was used to mark the PCR duplicates with a REMOVE_DUPLICATES parameter setting of false. BaseRecalibrator generated

the recalibration table by the known sites dbSNP and 1000G VCF resources. ApplyBQSR outputted the recalibrated final BAM files for HaplotypeCaller. In our analysis, the --contamination-fraction-to-filter parameter was set to a fixed value of 0.05; however, the optimal way to use this value is to supply each HaplotypeCaller task with the value appropriate for each sample individually. We first obtained the GVCF file for each sample and combined all the GVCF files into a single VCF files using GATK4 GenomicsDBImport and GenotypeGVCFs following the suggested pipelines. VariantFiltration was used for filtering excessively heterozygous variants (ExcessHet >54.69) marked with ExcessHet. We calculated the VQSLOD value of each SNV and INDEL variant by setting the max-Gaussians value to 5 and annotating with ReadPosRankSum, MQRankSum, DP, QD, FS and SOR for SNVs, with ReadPosRankSum, DP, QD and FS for INDELs. In the ApplyVQSR filtration, 99.0 was applied for the truth sensitivity level for INDELs and 99.6 for SNPs. All the passed variants were retained for the downstream analyses. We also identified the variants based on the reference genome GRCh37 referring to the same pipeline to provide the allele frequency of variants and reference panel.

For the X chromosome, we called the genotype in the pseudo-autosomal region (PAR) and non-pseudo-autosomal region (non-PAR), separately. We defined the parameter ploidy with 2 for females and 1 for males in non-PAR, while the parameter was 2 for all individuals in PAR. Then we merged the data together. We only called the genotypes on the Y chromosome for males as the haploid chromosome.

**Sample and variant filtrations**. The sex was inferred by the ratio of homozygous and heterozygous SNP variant in chromosome X. Females have homozygote/heterozygote ratio less than 4. In comparison to self-reported sex, 36 samples were not consistent and the mismatch rate was 0.8%. The FREEMIX scores were used to estimate DNA contamination by verifyBamID version 1.1.3 with --maxDepth 100 --precise --minMapQ 20 --minQ 20 --maxQ 100 with the allele frequencies inferred from our SNP genotyping array data[75]. In total, 15 samples with FREE-MIX scores >0.05 were excluded. We identified the duplicates samples by KING version 2.2.4 --duplicate with default values and by Plink version 1.9 --genome with the proportion of IBD > 0.95 (PI_HAT > 0.95)[76] removed 40 duplicated individuals or MZ twins[77]. Finally, 4480 samples were retained in the final cohort.

In the raw calling set (GRCh38 build), 99,774,562 SNVs and INDELs on autosomes, 4,186,591 SNVs, and INDELs on sex chromosomes were identified. Of these, 1,067,527 variants were excluded with the value 54.69 of ExcessHet (Phred-scaled p value for the exact test of excess heterozygosity). At a truth sensitivity of 99.6% for SNVs and 99% for INDELs, 11,160,919 SNVs and 2,690,120 INDELs were removed. We used the bcftools filter to remove the 2,933,200 SNVs and 2,643,970 INDELs variants closer to INDEL with SnpGap 3 and IndelGap 5. Then we filtered 12,274 INDELs with a length of more than 50 bp. Individuals genotypes were changed to missing if the genotype quality score (GQ) <20. The missing and low confidence genotypes were refined using BEAGLE 5.1[78]. Lastly, we excluded 1,947,496 variants with a HWE p value <1E−06 by VCFtools (v 0.1.13)[79].

**Variant annotations**. The functional annotation of variants was performed with the ANNOVAR tool[25]. We annotated the gene name, protein change, location, and function for all the variants. The pathogenic or benign variants were annotated by SIFT[80], PolyPhen-2[81], MutationTaster[82], and ClinVar version 20210927[28]. We manually curated the variants according to the standard and guideline recommended by the American College of Medical Genetics and Genomics and the Association for Molecular Pathology[83].

**Genotyping and quality control**. The 5841 samples of the WBBC Project and 184 individuals (13.3×–54.7×, median = 17.1×) sequenced by WGS were genotyped by ASA BeadChip designed for the East Asian population. About 680k variants were genotyped successfully for each sample. In total, 636,342 bi-allelic SNV variants were retained in all the autosomal chromosomes. We then calculated the genetic relationship matrix across all individuals using variants with MAF > 0.01 by GCTA v1.91[84], and removed 339 cryptically related samples with the pairwise genetic relationship coefficient >0.025. The 18,789 variants and 7 samples with a missing call rate >5% were excluded by Plink version 1.9[76]. The 7903 variants deviating from Hardy–Weinberg equilibrium at p < 1E−06 were excluded. We computed the allele frequencies of all the variants for the verifyBamID analysis. Then, we further filtered 139,371 SNVs with the MAF < 0.01. Finally, 5679 individuals and 470,279 common bi-allelic autosome variants passed the filters and QC.

**Evaluation of genotype concordance**. We applied the sequencing and genotype data from 184 individuals (13.3×–54.7×) to estimate the WGS calling accuracy. The genotype from SNP arrays was considered the true genotype and our calling variants were test set. We extract the SNP genotyping array sites from 184 WGS samples. The variants with calling rates for each sample more than 95% and the frequency more than 1% were retained. Finally, 483,755 variants in autosomes detected by both WGS and SNP array were used to estimate the genotype concordance. We also conducted the LD-based genotype refinement via BEAGLE 5.1 with default settings[78]. We computed the heterozygote discordance, NR genotype concordance, specificity, and NR sensitivity for the shared variants (Supplementary Fig. 15)[85].

**Calculation of the singleton density score (SDS)**. We estimated the kinship of samples by and IBDkin[86] and KING --kinship[77]. In total, 29 samples in one of the pair individuals within the degree 3 of relationship or the kinship coefficient more than 0.0442 were removed. Finally, SDS analysis was conducted with 4334 Han Chinese samples (2405 healthy samples and 1929 Parkinson's disease samples). We extracted the bi-allelic SNVs in all autosomal variants and filtered the SNVs by Hardy–Weinberg equilibrium (p < 1E−06). We downloaded the Homo sapiens ancestral annotation information from the Ensembl release-98. SNVs without defined ancestral allele were subsequently removed. Additional SNPs were excluded by MAF < 5% and less than 5 individuals for each of the three genotypes. The final dataset included 4,258,941 SNVs and 17,951,337 singletons for the SDS computation.

Gamma-shape was estimated with Gravel_CHB as a demographic model for each DAF bin by 0.005 from 0.05 to 0.95. The haplotypes were set to twice the number of individuals. We excluded the centromeres and heterochromatic regions with chromosome boundaries files. The skip boundary missing singletons fraction threshold was 0.5. The raw SDS scores were computed using recommended scripts and standardized within each 1% bin of DAF for each chromosome by calculating z-scores. Two-tailed p values were calculated by whole-genome-wide standardized SDS z-scores.

**Reference panel construction**. The multi-allelic sites were split into bi-allelic sites via the BCFtools norm tool version 1.7[87]. We filtered the variants with --max-missing 0.9 and --hwe 0.000001 using VCFtools version 0.1.13[79]. BEAGLE version 5.1 was used to perform haplotype phasing of all 4480 samples with default settings. The chromosomes were divided into chunks of 1 Mb with 0.1 Mb overlapping. We conducted the haplotype re-phasing with SHAPEIT v2 by windows 0.5, states 200 and effective size 14,269[88]. Finally, the SHAPEIT haplotypes were converted into VCF format files.

**Pre-phasing and imputation**. We pre-phased the array dataset by SHAPEIT setting the effective-size parameter to 14,269 as the software recommended for the Asian population[89]. Imputation was then performed with our own haplotype reference panel, which consisted of 8978 haplotypes at 34,948,874 SNPs (no singleton), by MINIMAC v4[90]. The length of chunks was set to 20 Mb with a 4 Mb overlap between contiguous chunks for the imputation. We employed Rsq to control the quality of imputed results and filtered out variants with Rsq ≤ 0.95.

**Reference panel evaluation for imputation in the Chinese population**. We evaluated the accuracy of genotype imputation for five reference panels in the Chinese population. These panels included the most widely used panel, the 1KG[22], and the largest Chinese-specific panel CONVERGE[45], and our own WBBC panel, and two combined panels that merged the WBBC datasets with the 1KG and EAS, respectively. The 1KG panel was population-diverse while the CONVERGE was Chinese-specific. The sequencing depth of CONVERGE was actually low, only ~1.7×. The imputation accuracy of these panels was then compared with each other by three different metrics (Supplementary Fig. 15).

The 1KG Project reference panel (Phase 3, v5a) was downloaded from the ftp sites (http://ftp.1000genomes.ebi.ac.uk/vol1/ftp/release/), and the CONVERGE Project reference panel was downloaded from the European Variation Archive with PRJNA289433 (http://ftp.ebi.ac.uk/pub/databases/eva/PRJNA289433/). For the 1KG, CONVERGE, and WBBC reference panels, we split multi-allelic variants into multiple bi-allelic variants and removed singletons and doubletons (minor allele counts, MAC ≤ 2) by using BCFtools. Besides, there were 184 samples that were included in both the WGS and DNA array genotyping for the evaluation purpose. These samples were held-out from the current WBBC panel. Finally, we obtained 3,284,591 variants and 5008 haplotypes for the 1KG, 1,115,342 variants and 23,340 haplotypes for the CONVERGE, and 2,089,508 variants and 8592 haplotypes for the WBBC. Note that all the manipulations were conducted on chromosome 2[14,91]. For two combined reference panels, the WBBC + 1KG and WBBC + EAS, we employed the reciprocal imputation approach to implement the combination to preserve maximal variants[71]. The EAS dataset was directly extracted from the 1KG, and sites with MAC equals zero were removed subsequently. We reciprocally conducted imputation for the WBBC/1KG and WBBC/EAS, and then respectively excluded 2663 and 2142 INDELs with incompatible alleles in panels that could fail the next panel-merging. BCFtools was used to finally merge the reference panels[87]. Eventually, the WBBC + 1KG combined panel consisted of 13,600 haplotypes at 4,450,989 variants, with 917,784 variants shared by both panels. The WBBC + EAS combined panel consisted of 9600 haplotypes at 2,411,382 variants, between them, 849,281 variants were shared. We extracted chromosome 2 from our QCed chip array dataset and randomly masked one fifteenth SNPs[14,91]. A total of 5679 individuals were included and 2600 SNPs were masked for the next evaluation.

We transformed the format of five panels into M3VCF and performed genotype imputation by jointly using Minimac3/4[90]. The length of chunks for imputation was set to 20 MB with 4 MB overlapped between contiguous chunks. The accuracy of different reference panels was evaluated by three metrics. In the first one, the estimated value of the squared correlation between imputed genotypes and true, unobserved genotypes (i.e., Rsq)[90], was calculated based on the imputed dosage and produced with the imputation results by Minimac4. This value was also the most

commonly used metric. In this study, an imputed variant with the $Rsq \geq 0.8$ was considered as "well-imputed". For the comparison purpose, we extracted 729,958 imputed variants that were shared by the five panels. The variants were then grouped into nine MAF bins (<0.1%, 0.1–0.2%, 0.2–0.3%, 0.3–0.5%, 0.5–1%, 1–2%, 2–5%, 5–20% and 20–50%) to differentiate the detailed imputation performance for variants with different MAF, especially for low-frequency and rare variants, which are usually difficult to impute accurately. We obtained average $Rsq$ values from Minimac4 info files and counted the well-imputed variants in each MAF bin. The second metric was NR-allele concordance. The variants that had been masked in the beginning were imputed by different panels. We then calculated the NR-allele concordance between imputed genotypes and the original ones in chip array for each individual (imputed vs. array)[91]. To gain a better understanding of the distribution of the genotype concordance, we separated the NR alleles into homozygote and heterozygote. The third metric was similar to the second, but the NR-allele concordance was calculated between imputed genotypes and WGS genotypes by the samples that we hold-out (imputed vs. WGS). The definition of concordance and corresponding formula was specified in Supplementary Fig. 15[85].

**Genotype imputation server.** Using the WBBC Pilot WGS data and 1KG Phase 3 data[22], we developed a genotype imputation server for public use. We included the WBBC and 1KG reference panel in the server and re-constructed two combined panels, the WBBC + EAS and WBBC + 1KG. All panels were built in both GRCh37 and GRCh38 version, and singletons were excluded. MINIMAC v3[90] was used here to build genotype data in the M3VCF format to save the computational memory. We developed the pipeline in Python and Bash, and employed MySQL for the management of data. For the VCF-formatted array data uploaded by users, the validity of data would be checked first. Before the actual imputation, there were some basic filtering steps conducted by BCFtools[87], including removing all mismatched SNPs, monomorphism, and duplicate SNPs. The 1KG was used here as the allele reference. The next phasing and imputation were performed using SHAPEIT v2 and MINIMAC v4[89,90]. We specified a policy of data security to protect the user's data across the entire interaction process with the server. Also, we wrote a help manual and illustrated all processes of our pipeline to facilitate users. Detailed information could be found in our website (https://wbbc.westlake.edu.cn).

**PCA, ADMIXTURE, and effective population size inference.** We removed the variants in imputed dataset by $Rsq \leq 0.95$, and merged it with our sequencing dataset by GATK v4.1.4.0[92], resulting in 9996 individuals and 2,016,533 bi-allelic SNPs. We further merged the WBBC dataset with the 1KG Project. After filtering SNPs by MAF $\leq 0.01$, a total of 1,857,766 bi-allelic SNPs with 100% call rate were left for subsequent analysis. We noted that the participants of the WBBC Project mainly came from three provinces of China, including Jiangxi (23.8%), Shandong (26%), and Hunan (31.4%). To avoid the potential bias of oversampling certain provinces[93], we randomly extracted 150 samples from each of the three provinces. Finally, 2056 Han population individuals, 205 Minority population individuals (Tujia, Zhuang, Yi, Mongolian, etc.), and 2504 1KG individuals were included. We then performed PCA[94], ADMIXTURE[95], and inference of effective population size. Note that the minority population individuals were held-out from each province group.

We excluded the SNPs with HWE $p$ value < 1E−06, MAF < 0.05 and genotype missing >0.05 using the Plink software[76]. Then we performed the LD-based SNP pruning with --indep-pairwise 50 10 0.5. The final datasets had 338,275 bi-allelic SNPs for PCA and ADMIXTURE analyses. We used the smartpca command from the software EIGENSOFT (v6.1.4)[96] and calculated the components for the first ten PCs. PC1 and PC2 were selected for the genetic diversity comparison, which were plotted by in-house R scripts.

ADMIXTURE analysis was conducted with 2,056 Han individuals, 103 CHB (Han Chinese in Beijing, China), 105 CHS (Han Chinese South, China), 93 CDX (Chinese Dai in Xishuangbanna, China), 104 JPT (Japanese in Tokyo, Japan) and 99 KHV (Kinh in Ho Chi Minh City, Vietnam) individuals from combined dataset by ADMIXTURE version 1.3.0 using default parameters[97]. To obtain the optimal K value, we analyzed the ADMIXTURE with 10 random seeds for each K ranging from 2 to 8. The default five-fold cross-validation procedure was carried out to estimate prediction errors. The $K$ value with the highest log-likelihood was selected as the most probable model.

For the ancient pattern projection of the East Asia, the published genome data of 396 ancient individuals (40,000–300 BP) from Eight countries or regions, including Russia ($n = 66$)[47,98–100], Mongolia ($n = 99$)[47,101,102], China mainland ($n = 144$)[46,47,103–105], Taiwan ($n = 46$)[47], Japan ($n = 61$)[47,106,107], Laos ($n = 3$)[107], Thailand ($n = 9$)[107,108], and Vietnam ($n = 19$)[107,108] were downloaded from the Allen Ancient DNA Resource (https://reich.hms.harvard.edu/allen-ancient-dna-resource-aadr-downloadable-genotypes-present-day-and-ancient-dna-data, version 44.3). Moreover, the 95 representative whole-genome sequences of the modern individuals were selected in the WBBC and 1KG, which were comprised of the Inner Mongolian (IMG, $n = 5$), North Han ($n = 10$), Central Han ($n = 10$), South Han ($n = 10$), and Lingnan Han ($n = 10$) from the WBBC, and 10 samples of each CHB, CHS, CDX, JPT and KHV from the 1KG[22]. PCA was performed by smartPCA program in the EIGENSOFT.v.6.1.4[96]. We used VCFtools v 0.1.13[79] software to calculate the pairwise $F_{ST}$ values within modern and ancient populations in China, Japan, Laos, Thailand, and Vietnam. We also estimated the

geographic distance among subgroups based on the latitude and longitude of samples.

We estimated the allele frequency trajectories for *ADH1A* (rs3819197), *ADH1B* (rs1229984), and *ALDH2* (rs671) genes with a hidden Markov model based on maximum-likelihood approach from time-series data of allele frequencies[44]. The R package "mathii/slattice" based on this method is available on GitHub (https://github.com/mathii/slattice). To perform this analysis, we selected 204 ancient samples (9500–300 BP) that covered SNP rs1229984 and 157 ancient samples that covered SNP rs3819197 from the 396 ancient individuals. We did not find ancient samples that covered SNP rs671. We also included 504 modern East Asian samples from 1000 Genome Project. We then divided ancient individuals into different generations (25 years per generation) according to their age and calculated DAF per generation (e.g., 9500–9475 years ago is the first generation contained 1 sample, 25 years ago to present is the 381st generation contained 504 modern samples). The allele frequency of each generation were then calculated as the input of the R package. We used the R packages "slattice" and defined the effective population size (Ne) as 10,000 and chose "Soft EM" Model in the estimation.

We further estimated the history of effective population size for four regions using SMC++[49]. Using the ancestral components analyzed by ADMIXTURE with $K = 4$, we designated 10 most representative samples with the high sequence-depth as the distinguished lineage sample for each region. We followed the suggestion of SMC++ authors and masked all low-complexity regions of the genome using the 1KG Phase 3 supported data[22], and kept all left bi-allelic SNPs for the next analysis. For each region, we repeated SMC++ 10 times according to each distinguished lineage sample. The combined results were used to form the composite likelihood for the final estimation. The per-generation mutation rate was set at 1.25E−8 and a generation time of 29 years was used to convert coalescent scaling to calendar time[14,49].

**$F_{ST}$ statistics, IBD analysis, and genetic drift estimation.** We next performed $F_{ST}$ statistics[109], genetic drift estimation, and IBD analysis. We calculated weighted Weir–Cockerham $F_{ST}$ estimates for each pair of the WBBC provinces and 1KG populations using VCFtools v 0.1.13[79] based on 1,857,766 bi-allelic SNPs. The window size was set to 50,000 and step size to 5000. We built $F_{ST}$ values matrix and performed hierarchical clustering with it using complete-linkage method implemented in the *hclust* function in the pheatmap package in R.

The IBD analysis was based on haplotypes of individuals. The genome-wide IBD segments were identified for all pairwise Han Chinese from 27 administrative divisions of China using Refined IBD software[110] with default settings. We built the IBD counts matrix for each pair of administrative divisions. Given that the sample size of 27 administrative divisions were different, we normalized the total IBD counts by sample size. For the IBD segment counts within administrative divisions (e.g., province "A"), $IBD_{normalized\ counts\ of\ A} = IBD_{total\ counts\ of\ A} / comb(N_A)$, where comb was the combination function in math and $N_A$ was the sample size of province "A". For the IBD segment counts between two administrative divisions (e.g., province "A" and "B"), $IBD_{normalized\ counts\ of\ A\ vs.\ B} = IBD_{total\ counts\ of\ A\ vs.\ B} / N_A \times N_B$, where $N_A$ and $N_B$ were the sample size of province "A" and "B", respectively. The hierarchical clustering was then performed based on the matrix by using the same method as $F_{ST}$ clustering.

We computed relative genetic drift estimates between each province using TreeMix v1.13 with default settings on the same SNPs as the $F_{ST}$ analysis used[50]. The genetic drift was represented by a "parameter" in TreeMix[50]. A maximum-likelihood tree for the Han Chinese population from 27 administrative divisions was then plotted. Note that the Heilongjiang province, which was located in the northernmost part of China, was set as the reference point. For judging the confidence in our tree topology, ten bootstrap replicates were generated by setting the -bootstrap -k flag ranging from 10 to 100 (step size = 10) to resample blocks of contiguous SNPs for drift parameter estimation. Plink version 1.9 was used in this part to calculate allele counts of SNPs for reformatting of input data that the software required[76].

**Calculation of iHS values.** To detect the genomic signatures of recent positive selection, we computed the iHS using the R package rehh v3.1.0[52,111]. The data from 2860 North, 148 Central, 5274 South and 92 Lingnan Han Chinese individuals were extracted from the imputed and phased files. Averagely, 1,925,157 bi-allelic SNVs were obtained in all autosome chromosomes in four Han populations. The SNVs were further filtered by Hardy–Weinberg equilibrium (--hwe 0.000001) and minor allele frequency (--maf 0.01) using the Plink software[76]. The ancestral allele of SNVs was defined by the data downloaded from Ensembl release-98. We removed SNVs without an ancestral allele state.

In total, 1,725,164 SNVs in North population, 1,712,580 SNVs in Central population, 1,720,051 SNVs in South population and 1,685,839 SNVs in Lingnan population passed quality control and were retained for statistical analysis. We performed iHS statistics independently for the population. The absolute values of the iHS scores were taken to analyze the data. We calculated the fraction of SNVs with $|iHS| > 2$ in 200 kb non-overlapping genomic windows ($N_{|iHS|>2} / N_{total}$) and excluded the regions with <20 SNVs[112]. The genes or genomic regions were defined within 100 kb of the identified non-overlapping SNVs. We performed the GO and KEGG pathway enrichment analysis for the adaptive candidate genes using R package clusterProfiler v3.16.0[113].

**Reporting summary**. Further information on research design is available in the Nature Research Reporting Summary linked to this article.

## Data availability

The sequencing and vcf data generated in this study have been deposited in the Genome Sequence Archive (GSA)[114] in National Genomics Data Center[115], China National Center for Bioinformation/Beijing Institute of Genomics, Chinese Academy of Sciences, under accession number HRA001385 (https://ngdc.cncb.ac.cn/gsa-human/). The Fastq data are available under restricted access for privacy protection and access can be obtained by application on the website. The user can register and login to the GSA database website (https://ngdc.cncb.ac.cn/gsa-human/) and follow the guidance of "Request Data" to request the data step by step. The access authority can be obtained for Research Use Only. The user can also contact the corresponding author directly. The processed frequency data are available at the WBBC website (https://wbbc.westlake.edu.cn/genotype.html). The online imputation service could also be available at the WBBC website (https://imputationserver.westlake.edu.cn/). Other data generated in this study are provided in the Supplementary Information/Source Data file. Source Data are provided with this paper. The data used in this study are the reference genome GRCh38 (https://gatk.broadinstitute.org/hc/en-us/articles/360035890811-Resource-bundle), 1000 Genome Project (http://ftp.1000genomes.ebi.ac.uk/vol1/ftp/release/), gnomAD (http://www.gnomad-sg.org/downloads), UK10K (https://www.uk10k.org/data.html), CONVERGE Project (http://ftp.ebi.ac.uk/pub/databases/eva/PRJNA289433/), dbSNP Build 151 (https://hgdownload.soe.ucsc.edu/downloads.html), and Allen Ancient DNA Resource (https://reich.hms.harvard.edu/allen-ancient-dna-resource-aadr-downloadable-genotypes-present-day-and-ancient-dna-data, version 44.3).

## Code availability

The scripts for R (version 4.0.2) and variant calling pipeline used in this study can be found in our GitHub repository (https://github.com/peikuan/WBBC).

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

## Acknowledgements

We thankfully acknowledge Kangyong Hu from the High-performance Computing Center at Westlake University and Bioinformatics Center at Xiangya Hospital for the computational support. We would like to thank Novogene Co., Ltd for their support and assistance in the genotyping of the study samples. This study was supported by the National Natural Science Foundation of China: 32061143019 (H.-F.Z.), and by the Westlake BioBank for Chinese (WBBC) funds from the Westlake University (H.-F.Z.), and by the National Key Plan for Scientific Research and Development of China: 2016YFC1306000 (B.-S.T.).

## Author contributions

H.-F.Z. conceptualized and designed the study. P.-K.C., W.-Y.B., M.-Y.Y., and S.K. conducted the data analysis. S.-H.Y., W.-W.Z., Y.S., and J.-Q.L. conducted the whole sequencing experiments. B.-S.T. and J.-C.L. provided the whole sequencing data from Hunan province. X.-W.Z., P.-P.Z., J.-W.X., P.-L.G., J.-G.T., Y.Q., G.T., S.-Y.X., L.X., P.-Y.W., M.-C.Q., and K.-Q.L. contributed to the collection of study samples. Y.-H.L., W.-Y.B., P.-K.C., S.-R.G., and N.L. designed the online website resource. H.-F.Z., P.-K.C., and W.-Y.B. drafted the manuscript. H.-F.Z., B.-S.T., J.-C.L., and S.K. reviewed and edited the manuscript. All authors contributed, discussed, and approved the manuscript.

## Competing interests

S.-H.Y., W.-W.Z., Y.S., and J.-Q.L. are employees of KingMed Diagnostics Co., Ltd. The other authors have no conflict of interest to declare.
