## [Peer Review File · Nature Communications]

Genomic analyses of 10,376 individuals in the Westlake BioBank for Chinese (WBBC) pilot projectReviewers' Comments:

Reviewer #1:

Remarks to the Author:

I am glad to see the majority of my previous concerns have been addressed in the revision. In general, I would very much like to see the publication of this paper because of the large amount of sequencing data it reports. The allele frequency data are now available online, but I am not sure if we can download and use the raw data.

"Raw sequencing data have been deposited to the CNGB Sequence Archive (CNSA) of China National GeneBank (CNGBdb) with accession number (CNP0001516) (<https://db.cngb.org/cnsa/>)". Can we download the data freely from this website after the publication of this paper or do we still need to submit an application for approval?

I still find the population genetic analyses are not that informative. The authors want to show and actually describe so many results but have not found an interesting high-value question to address.

In the response, the authors stated "We hope that, in the future, the genetic studies for Chinese population can follow the standard in our study to classify the subgroup populations in China, this will facilitate the match of case-control in the association study in Chinese population." But I have not found the authors set any standard in the paper and I don't think there is a standard for population structure inference, since the structure largely depends on how many samples and markers you use for constructing the PCA and admixture plots. You find north, central, south, and Lingnan subgroups, ok, that's fine, but please don't take it as a standard.

Reviewer #2:

Remarks to the Author:

The authors present a new public genomic resource for the Chinese population, along with a web interface to the data, and an imputation server leveraging the new population. The results show that this powerful new resource for genotype imputation far surpasses the currently available reference panels. A variety of population genetics results can also be derived from the data, including effective population size for some of the geographic regions of China. These data are also used to correct and refine the current understanding of Chinese population structure. Allele frequencies of pathogenic variants in CLINVAR are compared between this cohort and previous studies on European cohorts, showing a significant differences in the gene related to erythropoietic protoporphyria. Recent allele frequency changes were also inferred, confirming some known sites of positive selection and new signatures in the SNX29 gene. Selection in the genes DNAH1 and WDR1 are described as having population-specific effects.

This manuscript is very clearly written, with the exception of a few cases of awkward phrasing. It has the most comprehensive and detailed methods section I've ever seen. The accessibility of the public resources is excellent. I have a few concerns about the data QC, specifically the low number of indels per individual genome and the high percentage of missense variants in the full dataset. If these values can be justified with a citation, that would alleviate my concern. My other comments just ask for some clarifications in the main text and suggestions for figure improvements:

Line 70 -- either? I would say progress has been make on both fronts, though not necessarily by the same researchers

Line 88 -- "was" to "is" -- this continues to be a problem, yes? Even with the new data presented here?

Line 115 -- very clear, thanks

Line 118 -- this Ts/Tv is for the post-filtering not raw, correct?

Line 194 -- is there a citation for the expected percentage? I would expect synonymous to be the highest proportion, not missense

Line 207 -- interesting finding re:FECH

Line 212 -- this number of SNPs per genome is about what I would expect, but the number of indels seems low -- typically I'm used to 8-10X more SNPs than indels, but this is closer to 15X

Line 220 -- given the low allele frequency, we would expect a high heterozygote to homozygote ratio according to Hardy-Weinberg

Line 228 -- is this the het/hom ratio per sample (i.e. across all the variants in an individual) or per variant (i.e. across all samples)?

Line 260 -- what does unreported mean?

Line 318 -- also an interesting finding and probably bears repeating in the conclusion as a recommendation for future studies

Line 380 -- does this mean there was no gene flow between central and north? I'm not sure what "closed" means here.

Line 474 -- what particularly is peculiar about the Qinghai Han Chinese?

Line 705 -- how did you utilize both references?

Line 730 -- did you call on chrX twice? It seems like you would have to call all samples diploid first, then infer sex assignments, then recall males as haploid in the non-PAR.

Line 736 -- did you supply the contamination FREEMIX values to HaplotypeCaller?

Line 749 -- does the ~1M variants filtered due to HWE include those already filtered in the GATK pipeline? The ExcessHet threshold should correspond to a p-value of $3.4e-06$ (https://github.com/broadinstitute/warp/blob/62e06b060b11e246c2623bd68351b2d15e3808bd/pipelines/broad/dna_seq/germline/joint_genotyping/JointGenotyping.wdl#L51), so that is a surprising number of remaining variants that are not in HWE.

Line 767 -- generating both types of data for a subset of individuals is a nice experimental design

Line 814 -- didn't you already use the KING results to remove related individuals?

Line 895 -- Shell → Bash? Or Bourne shell (sh)?

Line 954 -- typo: ADMIXTURE

Supp fig 1e -- this is very hard to interpret with so many colors. It would be helpful to annotate the largest slices with their categories, i.e. 54.22% missense

Supp fig 4 -- legend is incomplete: Can you reiterate the administrative divisions or refer to the text for supp fig 5? Or maybe this plot can use dots instead of letters since the breakdown of administrative divisions is in the following plot.

Reviewer #3:

Remarks to the Author:

Title: "Genomic analyses of 10,376 individuals in the Westlake BioBank for Chinese (WBBC) pilot project"

Authors: Cong PK, Bai WY, Li JC, ..., Tang BS, and Zheng HF

The manuscript describes WGS (13x) of 4,535 genomes and array-based genotyping of 6,025 individuals. Using singleton density score (SDS) analysis of WGS data, the authors recapitulated known loci in ADH cluster, ALDH2, and HLA, and identified novel loci in SNX29, and suggestive signals in DNAH1 and WDR1. As expected, imputation accuracy has improved compared to 1000G, and became higher when combined. Imputation server is provided as a community resource. Population genetic analysis revealed clusters of regions with higher genetic similarity than others.

The amount of resource invested for this study is impressive. The findings from SDS analysis is potentially interesting. The availability of imputation server would be a useful addition to the community.

However, the manuscript have at least three major issues to be addressed with a substantial revision.

1. There is not much new scientific insight provided except for the identification of SNX29 loci (which might be results of confounding of oversampling Parkinson's disease). This is a little bit disappointing considering the ample and unique resource that authors have access to. Many scientific and technical claims in the manuscript is not new or surprising, but rather recapitulated typical expectation from genetic studies at this scale.
2. Some of the scientific claims made in the manuscript lacks important details or appears not quite credible. There is no detailed description of methods on how such estimates were obtained, including the age when the selective pressure was strong. The authors need to add much more detailed support and reasoning on the population genetic claims made in this manuscript.
3. There are serious level of inconsistencies or unexpected numbers the description of technical details (e.g. genome build for alignment, QC of arrays). These lack of details question the reliability and the quality of the resource and the scientific findings.

I will elaborate these concerns across the manuscript, indicating whether each issue is major or minor issues to address

Major / Line 50 : "We found that 5.05% of the rare variants in WBBC were common in European population, and some trait-associated common variants in European had much lower allele frequency in Chinese" - This descriptive statistics does not reflect particular scientific value. If the authors want to claim that such numbers are significantly higher or lower compared to expectation from other population, they need to be shown rigorously.

Major / Line 57 : "and the selection of the alcohol metabolism genes (ADH1A and ADH1B) strengthened from about 4,000 years ago in East Asia". This claim does not have a credible basis. It was never described how these estimates were obtained throughout the manuscript.

Minor / Line 66 : "Genetic divergence from north to south was more noticeable in the ancient than

modern populations." This qualitative statement does not add much scientific value. Providing a quantitative statement to describe how ancient vs modern divergence differs compared to expectation would be informative enough for abstract.

Major / Line 122 : "Additionally, C>T, G>A, A>G and T>C constituted the most common SNV substitution types and the length of INDELS mainly distributed between -10bp and 10bp " - this is so obvious and not worth stating this way.

Minor / Line 124 : "We provided a database of genetic variations for the Han population in four sub-regions (North, Central, South and Lingnan)". This is not just a database but a web application that enables lookup of variants.

Minor / Line 128 : "45,696,726 variants were found not to present in the 1000 Genome Project (1KG), gnomAD and UK10K (Fig. 1c)", shouldn't it be "or", not "and"?

Major / Line 147-186 : The entire section of "trait-associated variants between populations" just provides descriptive statistics on the variants with large differences in DAF without any statistical or population genetic theory behind it, so it does not really provide a scientific value. Please either completely rewrite the section that enables proper population genetic interpretation, or remove the section from the manuscript.

Major / Line 188-209 : Variant annotation section provides descriptive statistics that do not differentiate this resource with other resources. When presenting numbers, please be concise, and interpret whether this is expected or unexpected (or statistically significant) compared to other similar resources. Without such effort, the presented numbers or examples do not provide much scientific value.

Major / Line 211-218 : This section provides an interesting number but it does not tell how it is similar or different compared to other resources (line 210-214). For disease-associated variants, the most important aspect is how genomic disparities affects the identification of pathogenic variants but such aspects were not considered.

Major / Line 218-222 : "In these pathogenic and deleterious variants, we observed the higher ratio of heterozygote and non-reference homozygote (Het/Hom) (Supplementary Table 7). The proportions of Het/Hom were also very high in novel SNVs and INDELS variants, which indicated that the majority of novel variants occurred as heterozygotes in the Han Chinese population". This is totally misleading. When matched by allele frequencies, this effect should be gone, and it is well known that pathogenic variants tend to be rare. Please remove this part from the manuscript.

Minor / Line 224-225 : "The homozygous pathogenic variants are responsible for recessive Mendelian disorders" - this is not a relevant introductory text of the paragraph and should be removed or modified to fit into the context.

Major / Line 225-235 : Analysis of pathogenic variants lacks the point. Are they really pathogenic, or are they false positives occurring due to genomic disparities? How different are those numbers compared to other populations? These should be the main questions, not descriptive statistics of those variants.

Major / Line 237 : One of the major concern with SDS analysis is that the individuals are highly concentrated into regions (Hunan and Jiangxi) but the impact of it is not very well discussed. How do the result change if a selected subset of individuals were analyzed, similar to other popgen analysis? Any possible artifacts associated with confounding with biased sampling procedure (and different distribution between WGS and arrays) must be evaluated more systematically.

Minor / Line 256 : To claim WDR1 as significant, there needs to be an additional justification on how significance threshold was determined.

Major / Line 366-442 : This section is unnecessarily long and does not provide convincing evidence to support the hypothesis. PCs can be indirect evidence but cannot be the major evidence to draw the conclusion, but the arguments primarily focused on subtle aspects on PCs. ADMIXTURE analysis does not support the hypothesis clearly either. Combining with 1000G also does not support the hypothesis clearly. Analysis of Allen ancestral DNA data may shed some light, but the analysis is not quantitative enough to support the claim strongly. Please re-analyze focusing on the hypothesis, removing irrelevant pieces of information.

Major / Line 705 : "The reads in each lane were aligned to the GRCh38 and GRCh37 human reference genome via the BWA mem tool to produce the SAM files, respectively." This statement very much concerns me. The authors claims that the reads were mapped to two genome builds. Then are there two variant callsets? No, there is only one variant callset (if there were two, both should have been described). The authors do not seem to fully understand the technical details of this misleading statement. It is possible that the reads were mapped into two reference genomes, but only the piece relevant to the manuscript should have been described. How were the reads exactly mapped? How was the reference genome built? Were any decoy sequences or HLA alleles were added? What was the exactly reference fasta file used (is it one available publicly in 1000G?). The authors also should have described how the variant calls were made on which genome build and how they were lifted over. The main analysis results must be based on a specific genome build and that has to be described. It is also not described whether specific centromere or repeat regions were excluded from analysis or not. These pieces of details are important but were not sufficiently articulated.

Minor / Line 743 : "Of these, 1,067,527 variants were excluded by ExcessHet.". More details need to be provided how "ExcessHet" is exactly defined.

Major / Line 762 : ASA-array has 738,980 variants but only 484,554 SNPs left after QC. This looks like a huge loss of variants and potentially raise a concern of over-filtering or poor data quality. The authors should describe a detailed statistics of the number of variants remove/retained in each QC steps, so that the readers can understand where the large chunk of loss of SNPs happened. Did it lose a lot of SNPs because they are monomorphic? Or did they fail HWE unexpectedly too many? This is very important to know.

Minor / 775: disconcordance => discordance

Major / Line 783 : The method section described that SDS was perform with 2,405 healthy or non-Parkinsons (normal is not an adequate wording) individuals and 1,929 Parkinsons disease. This may play as a confounding effect, and it is important to make sure that the results of SDS analysis are not affected by the disease status. Since SNX29 is reported to be associated with Parkinson's disease, this is particularly important to address (then SNX29 selection claim should be nullified).

Minor / Line 817 and 921, Supp Figure - HWE threshold are described inconsisgtently (1×10^{-6} vs $10e^{-6}$). Please make it consistent.

Title: Genomic analyses of 10,376 individuals in the Westlake BioBank for Chinese (WBBC) pilot project

Thank you for the opportunity to resubmit our manuscript to *Nature Communications* for consideration for publication. For ease of reading we have directly pasted the Reviewer's comments below, which are in Calibri font. **Our page and line number references refer to the Marked version of the manuscript.**

Our responses are in Times New Roman font, preceded by “**Response:**”

Reviewer #1 (Remarks to the Author):

I am glad to see the majority of my previous concerns have been addressed in the revision. In general, I would very much like to see the publication of this paper because of the large amount of sequencing data it reports. The allele frequency data are now available online, but I am not sure if we can download and use the raw data. “Raw sequencing data have been deposited to the CNGB Sequence Archive (CNSA) of China National GeneBank (CNGBdb) with accession number (CNP0001516) (<https://db.cngb.org/cnsa/>)”. Can we download the data freely from this website after the publication of this paper or do we still need to submit an application for approval?

Response: Thanks so much for your interests and affirmation to our work. We have deposited our raw sequencing data into Genome Sequence Archive (GSA) under the China National Center for Bioinformation (CNCB) and National Genomics Data Center (NGDC). The data are now publicly accessible at <https://ngdc.cncb.ac.cn/gsa-human/> under accession number HRA001385. GSA is one of repositories recommended by the journal. We have tried to release the raw data freely, however, according to the regulations of the Human Genetic Resources Administration of China (HGRAC), our data should be controlled-access. Researchers

could get permission to access to the raw genetic data through the GSA database. The allele frequency of variants could be downloaded freely from our website. Please see “Data Availability” on Line 1139, Page 53.

I still find the population genetic analyses are not that informative. The authors want to show and actually describe so many results but have not found an interesting high-value question to address.

Response: We agree that the “North-South” population structure of Chinese Han were reported in previous studies, However, our findings should provide some additional insights into the population structure of Chinese Han. The PCA analysis, the F_{st} , the IBD analysis and genetic drift estimation all support that the geographical boundaries of the Qinling-Huaihe Line and Nanling Mountains separated the Chinese Han into four Subgroup. We also added some quantitative statement for the genetic analyses on Line 420 and Line 471. We further investigated the demographic history and signatures of recent positive selection in four sub-regions. And we hope our findings could be helpful to the case-control design of association study in the Chinese population. We have discussed this on Page35, Line 734.

In the response, the authors stated “We hope that, in the future, the genetic studies for Chinese population can follow the standard in our study to classify the subgroup populations in China, this will facilitate the match of case-control in the association study in Chinese population.” But I have not found the authors set any standard in the paper and I don’t think there is a standard for population structure inference, since the structure largely depends on how many samples and markers you use for constructing the PCA and admixture plots. You find north, central, south, and Lingnan subgroups, ok, that’s fine, but please don’t take it as a standard.

Response: Sorry for the misunderstanding, you are right that we are not setting the standard for the population structure inference, what we want to say is that, we hope

our findings could be helpful to the case-control design of association study in the Chinese population. We have added this to Discussion. Page 35, Line 734.

Reviewer #2 (Remarks to the Author):

The authors present a new public genomic resource for the Chinese population, along with a web interface to the data, and an imputation server leveraging the new population. The results show that this powerful new resource for genotype imputation far surpasses the currently available reference panels. A variety of population genetics results can also be derived from the data, including effective population size for some of the geographic regions of China. These data are also used to correct and refine the current understanding of Chinese population structure. Allele frequencies of pathogenic variants in CLINVAR are compared between this cohort and previous studies on European cohorts, showing a significant differences in the gene related to erythropoietic protoporphyria. Recent allele frequency changes were also inferred, confirming some known sites of positive selection and new signatures in the SNX29 gene. Selection in the genes DNAH1 and WDR1 are described as having population-specific effects.

This manuscript is very clearly written, with the exception of a few cases of awkward phrasing. It has the most comprehensive and detailed methods section I've ever seen. The accessibility of the public resources is excellent. I have a few concerns about the data QC, specifically the low number of indels per individual genome and the high percentage of missense variants in the full dataset. If these values can be justified with a citation, that would alleviate my concern. My other comments just ask for some clarifications in the main text and suggestions for figure improvements:

Line 70 -- either? I would say progress has been make on both fronts, though not necessarily by the same researchers

Response: Thanks, we have revised this sentence accordingly (Page 5, Line 72).

“Over the past decade, great progress has been made to unravel the genetic basis of complex traits/diseases and the human evolutionary history.”

Line 88 -- “was” to “is” -- this continues to be a problem, yes? Even with the new data presented here?

Response: Thanks, we have made changes accordingly. (Page 5, Line 89)

Line 115 -- very clear, thanks

Response: Thanks.

Line 118 -- this Ts/Tv is for the post-filtering not raw, correct?

Response: Yes. It is post-filtering, not raw. To avoid misunderstanding, we revised this sentence. “Here, we identified 81,498,995 variants ($Ts/Tv = 2.15$) after filtration from 103.96 million total raw variants.” (Page 7, Line 117)

Line 194 -- is there a citation for the expected percentage? I would expect synonymous to be the highest proportion, not missense

Response: In our data, for the common variants ($MAF > 5\%$), the number of synonymous variants is larger than the number of missense variants, but for the lower frequency and rare variants ($0.5\% \leq MAF \leq 5\%$ and $MAF < 0.5\%$), the number of missense variants is larger than synonymous variants (Supplementary Table 3). Most of novel variants had lower allele frequency in WBBC. In the NARD project, they observed similar percentage that the number of missense/frameshift was higher than the silent/nonframeshift in total, lower frequency and rare variants (Table 1 in PMID: 31640730). And the number of synonymous variants was higher among common

variants. In the 1KJPN, the fraction of nonsynonymous SNPs was distinctly higher than synonymous for the very-rare variants (Figure. 2c in PMID: 26292667). In addition, we checked our novel variants, the number of missense variants were higher than the number of synonymous variants on each chromosome. Missense variants accounted for 54.22%, synonymous variants took up 29.69% and splicing sites made up 10.81% of the novel variants. (Page 10, Line 195 - Line 206)

Line 207 -- interesting finding re:FECH

Response: Yes, we found that the incidence of erythropoietic protoporphyria (EPP) varied significantly among countries, even in the European countries (0.03-0.36 per million). (PMID: 23114748)

Line 212 -- this number of SNPs per genome is about what I would expect, but the number of indels seems low -- typically I'm used to 8-10X more SNPs than indels, but this is closer to 15X

Response: Thanks so much for your careful reading. We found some bugs in our Perl script for the statistic of SNP and INDEL per genome. We fixed the bugs and re-analyzed the data. Now, we identified 257,832 INDELS per genome. We have updated the values in the main text and Supplementary Tables for each individual. (Page 12, Line 228 - Line 234). Comparing to the UK10K project (705,684 INDELS per genome), we have low number of INDELS each individual. However, comparing to SG10K (262,070 per genome) and NARD (0.3 M per genome), which had the similar sequencing depth, the number of INDELS was reasonable. The reason why we had low number of INDELS was the stringent approach for variant filtration. In fact, we ever used different threshold values and obtained more INDELS, but we got more false positive calls and the genotype imputation accuracy was reduced.

Line 220 -- given the low allele frequency, we would expect a high heterozygote to homozygote ratio according to Hardy-Weinberg

Response: Yes. The pathogenic and deleterious variants usually have low allele frequency. We observed the high heterozygote to homozygote ratio, consistent with the expected value according to Hardy-Weinberg. Now we have deleted this sentence (Page 13, Line 252). We have rewritten this section to compare the difference with other NGS projects. (Page 12, Line 228 - Line 245)

Line 228 -- is this the het/hom ratio per sample (i.e. across all the variants in an individual) or per variant (i.e. across all samples)?

Response: We used the number of all the heterozygous and homozygous variants in an individual to calculate the het/hom ratio, and then got the average value. Page 12, Line 237.

Line 260 -- what does unreported mean?

Response: What we wanted to say was that the top SNP rs78947691 and rs148629931 were not reported in the published literatures. The *DNAH1* gene and *WDR1* gene may associated with male infertility and gout development, respectively. To avoid misunderstanding, we removed this word and revised the sentence (Page 15, Line 295).

Line 318 -- also an interesting finding and probably bears repeating in the conclusion as a recommendation for future studies

Response: Thanks so much. In our study, the WBBC panel yielded substantial improvement for imputation accuracy for low-frequency and rare variants than existing panels. Comparing to the CONVERGE panel, we had less samples in the

WBBC; however, the higher coverage sequencing depth improved the genotype accuracy for variant calling. We emphasized the finding in the Discussion section. (Page 34, Line 715).

Line 380 -- does this mean there was no gene flow between central and north? I'm not sure what "closed" means here.

Response: Sorry for the misunderstanding, there was gene flow between central and north. Comparing to the central and south, the ancestral composition of the central was more similar to north. North and central Han had more gene exchange. This is actually a typo, we were trying to express "where Central Han were close to North" (Page 21, Line 425). We have corrected the word. Thanks for your careful reading.

Line 474 -- what particularly is peculiar about the Qinghai Han Chinese?

Response: Qinghai province is located in the high altitude region in China, the average altitude of whole province is greater than 3000 meters. For the most of North and South region of China, the altitudes are much lower than that. The high altitude could accompany cause some special environments, for example, the lower oxygen concentration. The genetic architecture of Qinghai Han Chinese could be special for adaption of such environment.

We added the sentence "Qinghai province is actually located in the high altitude region of China (average altitude > 3000 meters), the genetic structure of Qinghai Han Chinese might be adapted to the local environment." in the latest manuscript (Page 25, Line 523).

Actually, Qinghai is contiguous to another high altitude region in China, Tibet. The sample size of Tibet Han Chinese in our data, however, was too small to conduct the genetic structure analysis.

Line 705 -- how did you utilize both references?

Response: At the beginning, we called the variants based on the GRCh37 human reference genome. But we found that some variants could not be mapped by LiftOver when converting the variants from one reference genome to another build. So we called the variants again with the GRCh38 human reference genome. We utilized the variants data identified by the reference genome GRCh38 for all analyses in our study. We provided the allele frequency of variants and reference panels on both GRCh37 and GRCh38 on the website (<https://wbbc.westlake.edu.cn/>). (Page 37, Line 789)

Line 730 -- did you call on chrX twice? It seems like you would have to call all samples diploid first, then infer sex assignments, then recall males as haploid in the non-PAR.

Response: Yes, firstly, we call all samples diploid and infer the sex for each sample. For the X chromosome, we called the genotype in the pseudo-autosomal region (PAR) and non-pseudo-autosomal region (non-PAR), separately. We defined the parameter ploidy with 2 for females and 1 for males in non-PAR, while the parameter was 2 for all individuals in PAR. Then we merged the data together. (Page 37, Line 792)

Line 736 -- did you supply the contamination FREEMIX values to HaplotypeCaller?

Response: Yes. We set the parameter "--contamination-fraction-to-filter 0.05". We used the FREEMIX score 0.05 to exclude the contamination samples.

Line 749 -- does the ~1M variants filtered due to HWE include those already filtered in the GATK pipeline? The ExcessHet threshold should correspond to a p-value of 3.4e-06

<https://github.com/broadinstitute/warp/blob/62e06b060b11e246c2623bd68351b2>

d15e3808bd/pipelines/broad/dna_seq/germline/joint_genotyping/JointGenotyping_wdl#L51), so that is a surprising number of remaining variants that are not in HWE.

Response: Firstly, we excluded the 1,067,527 by ExcessHet. And before HWE filtering, we did one more step following the UK10K cohort's method (PMID 26367797), that we set the individual's genotypes to missing if the genotype quality score (GQ) <20 and conducted the genotype refinement to improve the genotype accuracy. The missing and low confidence genotypes were refined. That is the reason we removed another 1,947,496 variants with a HWE p value < 1E-06 by VCFtools. We added this to the Method section (Page 39, Line 818).

Line 767 -- generating both types of data for a subset of individuals is a nice experimental design

Response: Thanks a lot.

Line 814 -- didn't you already use the KING results to remove related individuals?

Response: Yes, we used the KING version 2.2.4 --duplicate with default values to remove 40 duplicate samples. (Page 38, Line 806). We also removed 339 cryptically related samples with the pairwise genetic relationship coefficient > 0.025 by GCTA v.191 (Page 39, Line 840)

Line 895 -- Shell → Bash? Or Bourne shell (sh)?

Response: It is "Bash". We have corrected it (Page 45, Line 976).

Line 954 -- typo: ADMIXTURE

Response: Thanks. We have corrected it (Page 49, Line 1047).

Supp fig 1e -- this is very hard to interpret with so many colors. It would be helpful to annotate the largest slices with their categories, i.e. 54.22% missense

Response: Thanks. We mainly focus on the distribution of SNVs and INDELS in the coding and splicing regions. Therefore, we merged the number of deletion/insertion, UTR and upstream/downstream etc (Fig. 1d and e).

Supp fig 4 -- legend is incomplete: Can you reiterate the administrative divisions or refer to the text for supp fig 5? Or maybe this plot can use dots instead of letters since the breakdown of administrative divisions is in the following plot.

Response: We have revised the legend and added the administrative divisions for each region (Page 74, Line 1494).

Reviewer #3 (Remarks to the Author):

Title: "Genomic analyses of 10,376 individuals in the Westlake BioBank for Chinese (WBBC) pilot project"

Authors: Cong PK, Bai WY, Li JC, ..., Tang BS, and Zheng HF

The manuscript describes WGS (13x) of 4,535 genomes and array-based genotyping of 6,025 individuals. Using singleton density score (SDS) analysis of WGS data, the authors recapitulated known loci in ADH cluster, ALDH2, and HLA, and identified novel loci in SNX29, and suggestive signals in DNAH1 and WDR1. As expected, imputation accuracy has improved compared to 1000G, and became higher when combined. Imputation server is provided as a community resource. Population genetic analysis revealed clusters of regions with higher genetic similarity than others.

The amount of resource invested for this study is impressive. The findings from SDS analysis is potentially interesting. The availability of imputation server would be a useful addition to the community.

However, the manuscript have at least three major issues to be addressed with a substantial revision.

1. There is not much new scientific insight provided except for the identification of SNX29 loci (which might be results of confounding of oversampling Parkinson's disease). This is a little bit disappointing considering the ample and unique resource that authors have access to. Many scientific and technical claims in the manuscript is not new or surprising, but rather recapitulated typical expectation from genetic studies at this scale.
2. Some of the scientific claims made in the manuscript lacks important details or appears not quite credible. There is no detailed description of methods on how such estimates were obtained, including the age when the selective pressure was strong. The authors need to add much more detailed support and reasoning on the population genetic claims made in this manuscript.

3. There are serious level of inconsistencies or unexpected numbers the description of technical details (e.g. genome build for alignment, QC of arrays). These lack of details question the reliability and the quality of the resource and the scientific findings.

Response: Thanks for these enlightening suggestions, we have responded accordingly below in detail.

I will elaborate these concerns across the manuscript, indicating whether each issue is major or minor issues to address

Major / Line 50 : "We found that 5.05% of the rare variants in WBBC were common in European population, and some trait-associated common variants in European had much lower allele frequency in Chinese" - This descriptive statistics does not reflect particular scientific value. If the authors want to claim that such numbers are significantly higher or lower compared to expectation from other population, they need to be shown rigorously.

Major / Line 147-186 : The entire section of "trait-associated variants between populations" just provides descriptive statistics on the variants with large differences in DAF without any statistical or population genetic theory behind it, so it does not really provide a scientific value. Please either completely rewrite the section that enables proper population genetic interpretation, or remove the section from the manuscript.

Response: These two questions were derived from the Results section “**Trait-associated Variants between Populations**”. We added this paragraph at first round of revision to compare the difference in the rare and low-frequency alleles between the Chinese (WBBC) and European in the NHGRI-EBI GWAS Catalog database, between WBBC and East Asian, South Asian, Admixed American, European and African in the 1000 Genome Project, and between WBBC and Inner

Mongolians, Koreans, Japanese, Vietnamese in the East Asian population. In consideration of the fact that the manuscript is already long and massive in content, we agree to remove the section from the manuscript. We also removed the sentence from the Abstract “We found that 5.05% of the rare variants in WBBC were common in European population, and some trait-associated common variants in European had much lower allele frequency in Chinese”. Thanks for your suggestion. (Page 8, Line 150 - Line 190).

Major / Line 57 : "and the selection of the alcohol metabolism genes (ADH1A and ADH1B) strengthened from about 4,000 years ago in East Asia". This claim does not have a credible basis. It was never described how these estimates were obtained throughout the manuscript.

Response: We have added the analysis in the Method section (Page 48, Line 1037 - Line 1044). We estimated the selection coefficient trajectories for *ADH1A* (rs3819197), *ADH1B* (rs1229984) and *ALDH2* (rs671) genes with a hidden Markov model using the ancient allele frequencies from East Asian ancient individuals (9,500 - 300 BP) and present-day allele frequencies in the WBBC. We divided ancient individuals into different generations (25 years per generation) according to their age and calculated derived allele frequency per generation. Allele frequency trajectories and selection coefficients for each gene were simulated by using a hidden Markov Model with an effective population size of 10,000. The derived allele of the SNP rs1229984 and rs3819197 was present around the 7,000 year ago, but was very rare for a long time (Fig. 2c). The strength of selection at the *ADH1A* (rs3819197) and *ADH1B* (rs1229984) gene increased in East Asian ancient individuals around 4,000 years ago.

Major / Line 122 : "Additionally, C>T, G>A, A>G and T>C constituted the most common SNV substitution types and the length of INDELs mainly distributed between -10bp and 10bp " - this is so obvious and not worth stating this way.

Response: Thanks for your suggestion. We have removed this sentence and deleted the Supplementary Figure in revision. (Page 7, Line 122 - Line 126)

Minor / Line 124 : "We provided a database of genetic variations for the Han population in four sub-regions (North, Central, South and Lingnan)". This is not just a database but a web application that enables lookup of variants.

Response: Thanks, we revised this sentence in Page 7, Line 126. "We provided a user-friendly website to search the annotation and allele frequency of genetic variants in the Chinese population, including the four Han sub-regions (North, Central, South and Lingnan)".

Minor / Line 128 : "45,696,726 variants were found not to present in the 1000 Genome Project (1KG), gnomAD and UK10K (Fig. 1c)", shouldn't it be "or", not "and"?

Response: Thanks for your careful reading, you are right, it should be "or", we have made changes accordingly. (Page 7, Line 132)

Major / Line 188-209 : Variant annotation section provides descriptive statistics that do not differentiate this resource with other resources. When presenting numbers, please be concise, and interpret whether this is expected or unexpected (or statistically significant) compared to other similar resources. Without such effort, the presented numbers or examples do not provide much scientific value.

Response: Thanks for your suggestion. As also requested by another reviewer, we now removed the descriptive numbers, and presented the findings from coding and splice regions and compared with Northeast Asian Reference Database (NARD) and Japanese population reference panel (1KJPN). For the common variants (MAF>5%),

the number of synonymous variants is larger than the number of missense variants. In contrast, the number of missense variants is larger than the number of synonymous in the lower frequency and rare variants ($0.5\% \leq \text{MAF} \leq 5\%$ and $\text{MAF} < 0.5\%$), especially for the rare variants (Supplementary Table 3). Most of novel variants had lower allele frequency in WBBC. In the NARD project, they observed similar percentage that the number of missense/frameshift was higher than the silent/nonframeshift in total, lower frequency and rare variants (Table 1 in PMID: 31640730). And the number of synonymous variants was higher among common variants. In the 1KJPN, the fraction of nonsynonymous SNPs was distinctly higher than synonymous for the very-rare variants (Figure. 2c in PMID: 26292667). In addition, we checked our novel variants, the number of missense variants were higher than the number of synonymous variants on each chromosome. Missense variants accounted for 54.22%, synonymous variants took up 29.69% and splicing sites made up 10.81% of the novel variants. (Page 10 Line 199 - Line 206)

Major / Line 218-222 : "In these pathogenic and deleterious variants, we observed the higher ratio of heterozygote and non-reference homozygote (Het/Hom) (Supplementary Table 7). The proportions of Het/Hom were also very high in novel SNVs and INDELS variants, which indicated that the majority of novel variants occurred as heterozygotes in the Han Chinese population". This is totally misleading. When matched by allele frequencies, this effect should be gone, and it is well known that pathogenic variants tend to be rare. Please remove this part from the manuscript.

Response: Thanks for your suggestion. We have removed two paragraphs of this part and rewrote the Results in latest version. We removed the descriptive statistics and compared our variants per individual to GoNL, UK10K, 1000 Genome Project and SG10K cohort (Page 12, Line 232 - Line 245).

Major / Line 211-218 : This section provides an interesting number but it does not tell how it is similar or different compared to other resources (line 210-214). For disease-associated variants, the most important aspect is how genomic disparities affects the identification of pathogenic variants but such aspects were not considered.

Minor / Line 224-225: "The homozygous pathogenic variants are responsible for recessive Mendelian disorders" - this is not a relevant introductory text of the paragraph and should be removed or modified to fit into the context.

Major / Line 225-235: Analysis of pathogenic variants lacks the point. Are they really pathogenic, or are they false positives occurring due to genomic disparities? How different are those numbers compared to other populations? These should be the main questions, not descriptive statistics of those variants.

Response: Thank you very much for your suggestion. We re-analyzed the data and rewrote this paragraph (Page 12, Line 234 - Line 245). We re-annotated the variants by the newly released version of ClinVar (v20210927) and checked the pathogenicity of each variant manually according to the standard and guideline recommended by the American College of Medical Genetics and Genomics (ACMG) and the Association for Molecular Pathology (AMP) (PMID: 25741868). In total, we identified 732 pathogenic or likely pathogenic variants in the healthy individuals. The majority of the candidate variants were associated with recessive hereditary diseases. We also compared the numbers and allele frequencies of variants with other cohorts. Each genome carried 3.6 ± 2.1 (mean \pm SD) pathogenic homozygote variants in Chinese Han population, which was consistent with the value (3.9 ± 2.0) in Chinese in the SG10K pilot study and fewer than the number of pathogenic homozygote variants in Malays (4.3 ± 2.1) and Indians (4.9 ± 2.2). We also identified some common pathogenic variants in East Asian population. However, these variants were not observed in the European population in the 1000 Genome Project.

Major / Line 237 : One of the major concern with SDS analysis is that the individuals are highly concentrated into regions (Hunan and Jiangxi) but the impact of it is not very well discussed. How do the result change if a selected subset of individuals were analyzed, similar to other popgen analysis? Any possible artifacts associated with confounding with biased sampling procedure (and different distribution between WGS and arrays) must be evaluated more systematically.

Response: We agree that the sample distribution in our study population was imbalanced, and most of the WGS samples were from human and Jiangxi. In the 4334 Han Chinese individuals, 3074 individuals are from Hunan province, 703 are from Jiangxi province and 557 individuals are from other provinces. To ease the comparison, we rerun the SDS analyses in five subgroups: 557 individuals in Hunan province (Hunan-557), 557 individuals in Jiangxi province (Jiangxi-557), 557 individuals in other provinces (Other-557), 703 individuals in Hunan province (Human-703) and 703 individuals in Jiangxi province (Jiangxi-703). Due to the insufficient samples, not all of the selection signals reported in the manuscript were significant in the subgroup analysis. The pattern between Hunan-557 and Jiangxi-557 was similar, and the *SNX29* was still significant in both Human-557 and Jiangxi-557 (as well as Human-703 and Jiangxi-703). Geographically, Hunan and Jiangxi are contiguous, belonging to the Southern region of China. The ancestral compositions of Hunan and Jiangxi showed similarity according to the results of PCA and admixture (supplementary Fig.7 and supplementary Fig. 9). The *ADH* gene cluster was significant in Other-557. MHC region and *ALDH2* were shown significance in Jiangxi-703. We discuss the limitation that only hundreds of samples were from other provinces. (Page31, Line 658 - Line 663).

For the calculation of *iHS* values, we used 8,374 Han individuals (2,860 North, 148 Central, 5,274 South and 92 Lingnan Han). 34.3 % of individuals came from the northern region of China. We observed that a significant enrichment of positively selected genes for ethanol metabolic process and ethanol oxidation in four sub-regions,

consistent with the selective signatures by whole genome-wide singleton density score (SDS) analysis in the Han Chinese population (Page 29, Line 607). We also observed intriguing enrichment of keratinocyte differentiation, epidermal cell differentiation and skin development only in the South and Lingnan Han. Although we detected the common signals, the region difference for positive selection existed in the Han Chinese population.

Minor / Line 256 : To claim WDR1 as significant, there needs to be an additional justification on how significance threshold was determined.

Response: In the legend of Fig 2. We defined the significance threshold. The horizontal red line indicates the significance threshold ($p < 5 \times 10^{-8}$). The P value of the top rs148629931 in *WDR1* gene was 5.44×10^{-8} . The value was very close to the significance threshold. We considered that the SNP rs148629931 was potential selection signal (Page 15, Line 291).

Minor / Line 66 : "Genetic divergence from north to south was more noticeable in the ancient than modern populations." This qualitative statement does not add much scientific value. Providing a quantitative statement to describe how ancient vs modern divergence differs compared to expectation would be informative enough for abstract.

Major / Line 366-442 : This section is unnecessarily long and does not provide convincing evidence to support the hypothesis. PCs can be indirect evidence but cannot be the major evidence to draw the conclusion, but the arguments primarily focused on subtle aspects on PCs. ADMIXTURE analysis does not support the hypothesis clearly either. Combining with 1000G also does not support the hypothesis clearly. Analysis of Allen ancestral DNA data may shed some light, but the analysis is not quantitative enough to support the claim strongly. Please re-analyze focusing on the hypothesis, removing irrelevant pieces of information.

Response: Thanks so much for your suggestion. To evaluate genetic differentiation within modern and ancient populations, we used VCFtools software to calculate the pairwise F_{st} values within modern and ancient populations, respectively (Page23, Line 471). The ancient population exhibited the larger genetic distances from 0.00365 to 0.25656 with median value of 0.03242, while modern population had the values from 0 to 0.01885 (Supplementary Fig. 10a). The genetic distance between northern and southern subgroups in modern Chinese population (0.00195) distinctly lower than the distance of 0.01902 in ancient Chinese population. Genetic divergence increased with geographic distance and was more noticeable from north to south in the ancient than modern populations. We revised the sentence in the abstract. "Genetic divergence

increased with geographic distance and was more noticeable in the ancient ($F_{ST} = 0.00365 \sim 0.25656$) than modern East Asians ($F_{ST} = 0 \sim 0.01885$).”

We removed some irrelevant sentences in the PCA and admixture sections and calculated pairwise F_{ST} values among subgroups (Page 20, Line 419 - Line 426). The average pairwise F_{ST} values within subgroups were 0.00024, 0.00023, 0.00037 and 0.00059 for North Han, Central Han, South Han and Lingnan Han, respectively. The pairwise F_{ST} between Central and North Han (0.00021) was much lower than the F_{ST} between Central and South Han (0.00083), between Central and Lingnan Han (0.00386), indicating that Central Han was overlapping with North and South Han, and more close to North Han. North-South Han (0.0015) and South-Lingnan Han (0.0013) had a relatively high F_{ST} values and were classified into different subgroups.

Major / Line 705 : "The reads in each lane were aligned to the GRCh38 and GRCh37 human reference genome via the BWA mem tool to produce the SAM files, respectively." This statement very much concerns me. The authors claims that the reads were mapped to two genome builds. Then are there two variant callsets? No, there is only one variant callset (if there were two, both should have been described). The authors do not seem to fully understand the technical details of this misleading statement. It is possible that the reads were mapped into two reference genomes, but only the piece relevant to the manuscript should have been described. How were the reads exactly mapped? How was the reference genome built? Were any decoy sequences or HLA alleles were added? What was the exactly reference fasta file used (is it one available publicly in 1000G?). The authors also should have described how the variant calls were made on which genome build and how they were lifted over. The main analysis results must be based on a specific genome build and that has to be described. It is also not described whether specific centromere or repeat regions were excluded from analysis or not. These pieces of details are important but were not sufficiently articulated.

Response: Sorry that we didn't make this clear. We did call the variants twice. At the beginning, we called the variants based on the GRCh37 human reference genome. But we found that some variants could not be mapped by LiftOver when converting the variants from one reference genome to another build. So we called the variants again with the GRCh38 human reference genome. We utilized the variants data identified by the reference genome GRCh38 for all analyses in our study. To ease the users, we provided the allele frequency of variants and reference panel on both GRCh37 and GRCh38 on the website (<https://wbbc.westlake.edu.cn/>). We revised the description in Page 36, Line 770 and Line 789.

We performed the variant calling by the GATK best practices pipeline. We used the reference genome files (human_g1k_v37_decoy.fasta and Homo_sapiens_assembly38.fasta) recommended by GATK, which can be download from the resource bundle (<ftp://gsapubftp-anonymous@ftp.broadinstitute.org/bundle/>). The reference genome contained the decoy sequences and HLA sequences. We added the description in the methods (Page 36, Line 771). In the raw calling set, we did not exclude the specific centromere or repeat regions. However, we excluded the relevant regions in the downstream analysis. For example, we excluded the centromeres and heterochromatic regions for SDS analysis (Page 41, Line 875) and performed the linkage disequilibrium for genetic structure analysis (Page 47, Line 1003).

Minor / Line 743 : "Of these, 1,067,527 variants were excluded by ExcessHet.". More details need to be provided how "ExcessHet" is exactly defined.

Response: ExcessHet is phred-scaled p-value for exact test of excess heterozygosity. This annotation estimates the probability of the called samples exhibiting excess heterozygosity with respect to the null hypothesis that the samples are unrelated. The higher the score, the higher the chance that the variant is a technical artifact or that there is consanguinity among the samples. ExcessHet is a parameter in the GATK pipeline. We revised this sentence (Page 38, Line 813). "Of these, 1,067,527 variants

were excluded with the value 54.69 of ExcessHet (Phred-scaled p-value for exact test of excess heterozygosity)".

Major / Line 762 : ASA-array has 738,980 variants but only 484,554 SNPs left after QC. This looks like a huge loss of variants and potentially raise a concern of over-filtering or poor data quality. The authors should describe a detailed statistics of the number of variants remove/retained in each QC steps, so that the readers can understand where the large chunk of loss of SNPs happened. Did it lose a lot of SNPs because they are monomorphic? Or did they fail HWE unexpectedly too many? This is very important to know.

Response: Thanks. We have revised the QC steps (Page 39, Line 835 - Line 845). We merged the genotyping and quality control into one paragraph. Illumina Asian Screening Array (ASA) was designed with 740K sites. About 680k variants were genotyped successfully for each sample. In total, 636,342 bi-allelic SNV variants were retained in all the autosomal chromosomes. We then calculated genetic relationship matrix across all individuals using variants with MAF > 0.01 by GCTA v1.91, and removed 339 cryptically related samples with the pairwise genetic relationship coefficient > 0.025. The 18,789 variants and 7 samples with a missing call rate > 5% were excluded by Plink version 1.9. The 7,903 variants deviating from Hardy-Weinberg equilibrium at $p < 1E-06$ were excluded. We computed the allele frequencies of all the variants for the verifyBamID analysis. Then, we further filtered 139,371 SNVs with the MAF < 0.01. Finally, 5,679 individuals and 470,279 common bi-allelic autosome variants passed the filters and QC.

The number of SNPs fail missingness rate or HWE seems normal, and the majority of SNPs filtered were with a small MAF (<0.01). Moreover, among them ~61K were monomorphic. For such monomorphic SNPs in our array dataset, we compared them with the East Asian (EAS) group in 1000G and our WBBC sequencing dataset, respectively. About 85.2% and 82.0% sites were found to be missing in EAS and our

sequencing data respectively, implying these SNPs designed in ASA-array could be extremely rare or hard to be fine-genotyped by sequencing.

Minor / 775: disconcordance => discordance

Response: Thanks. We have corrected the typo (Page 40, Line 856).

Major / Line 783: The method section described that SDS was perform with 2,405 healthy or non-Parkinsons (normal is not an adequate wording) individuals and 1,929 Parkinsons disease. This may play as a confounding effect, and it is important to make sure that the results of SDS analysis are not affected by the disease status. Since *SNX29* is reported to be associated with Parkinson's disease, this is particularly important to address (then *SNX29* selection claim should be nullified).

Response: Thanks. We replaced the “normal” with “healthy”. To make sure that the SDS results would not be affected by the disease (Parkinson’s disease), we rerun several SDS analyses within randomly selected “healthy” individuals at different sample size from 350 to 2100, with each analysis adding in 350 individuals (350, 700, 1050, 1400, 1750 and 2100 healthy individuals). The selection signals were not very significant in *SNX29* gene in the fewer number of individuals. However, we observed the sign of position selection in the top SNP on *SNX29* gene. We found the significant selection signatures of *SNX29* gene in 1050 healthy individuals, 1400 healthy individuals, 1750 healthy individuals, 2100 healthy individuals and all the individuals (2,405 healthy individuals and 1,929 Parkinson’s disease individuals). No difference in the *SNX29* gene were observed for the healthy and Parkinson’s disease individuals.

Previous study reported that *SNX29* was a new susceptibility gene associated with schizophrenia (SCZ), bipolar disorder (BPD) or major depressive disorder (MDD) (PMID: 33143498), In fact, we have an ongoing project of genome-wide association study (GWAS) with PD cases and controls, we did not observe the significant

association of *SNX29* with PD in that study. We added the description in Result (Page 14, Line 285) and Discussion section (Page 32, Line 665).

Minor / Line 817 and 921, Supp Figure - HWE threshold are described inconsistently (1×10^{-6} vs 10^{-6}). Please make it consistent

Response: Thanks. We have corrected the typo and used the $1E-06$. (Line 821, Line 842, Line 866 and Line 1002).

Reviewers' Comments:

Reviewer #1:

Remarks to the Author:

I suggest an acceptance of the paper, but also want the authors to write the Data availability clear in the main text.

In the main text, the authors stated:

"The raw sequence data reported in this paper have been deposited in the Genome Sequence Archive in National Genomics Data Center, China National Center for Bioinformation / Beijing Institute of Genomics, Chinese Academy of Sciences, under accession number HRA001385 that are publicly accessible at <https://ngdc.cncb.ac.cn/gsa-human>".

But in the response, the authors said:

"We have tried to release the raw data freely, however, according to the regulations of the Human Genetic Resources Administration of China (HGRAC), our data should be controlled-access".

I am very confused here. If the raw data are not publicly accessible, please make it clear in the main text to avoid any misunderstanding to the readers.

Reviewer #2:

Remarks to the Author:

Thank you for the hard work revising the manuscript. Most of the concerns in the previous revision have been addressed to my satisfaction.

As a minor note, with regards to our discussion of the use of the FREEMIX contamination estimate, the pipeline used here may be unfairly reducing the quality of the data. Supplying a constant value of 0.05 as the contamination value to HaplotypeCaller will lead to all variant calls effectively having 5% of their reads removed, which is roughly equivalent to reducing your sequencing coverage by 5%. The optimal way to use this value is to supply each HaplotypeCaller task with the value appropriate for each sample individually. I understand that making this change to the pipeline would require reanalyzing a large amount of data. There is nothing inaccurate about the method used here, but it's worth noting and pointing out that it's not quite in accordance with the GATK Best Practices.

Reviewer #3:

Remarks to the Author:

The authors properly addressed most of my comments. However, there is one outstanding comment that still remains and must be address before the publication.

Original comment:

Line 57 : "and the selection of the alcohol metabolism genes (ADH1A and ADH1B) strengthened from about 4,000 years ago in East Asia". This claim does not have a credible basis. It was never described how these estimates were obtained throughout the manuscript.

Response:

We have added the analysis in the Method section (Page 48, Line 1037 - Line 1044). We estimated the selection coefficient trajectories for ADH1A (rs3819197), ADH1B (rs1229984) and ALDH2 (rs671) genes with a hidden Markov model using the ancient allele frequencies from East Asian ancient

individuals (9,500 - 300 BP) and present-day allele frequencies in the WBBC. We divided ancient individuals into different generations (25 years per generation) according to their age and calculated derived allele frequency per generation. Allele frequency trajectories and selection coefficients for each gene were simulated by using a hidden Markov Model with an effective population size of 10,000. The derived allele of the SNP rs1229984 and rs3819197 was present around the 7,000 year ago, but was very rare for a long time (Fig. 2c). The strength of selection at the ADH1A (rs3819197) and ADH1B (rs1229984) gene increased in East Asian ancient individuals around 4,000 years ago.

New comment:

The description of the method is not detailed enough. The method is not published elsewhere, and the details of HMM is not clearly stated enough so I am unable to judge how credible the inference is. It is very hard to specify the exact year when the selection happened especially when the ancient genomes are not from the same region and when the ancient genomes are of limited sample size. Much more detailed description of the algorithm is necessary to justify the conclusion, and I suggest to make a more simplistic conclusion or provide additional mathematical details (probably multiple pages) how the inference was made in a enough detail that the method and conclusion can be fairly evaluated and can be reproduced elsewhere.

The results that ADH1A and ADH1B was under strong selection is further questions based on the new results of SDS stratified by province. In that result, chr4 has barely any signals except for 1 other province. This suggest that population structure may played a role in the spurious selection signals. More sophisticated method is warranted to disentangled the selection effect from population structure between current and ancient genomes.

Title: Genomic analyses of 10,376 individuals in the Westlake BioBank for Chinese (WBBC) pilot project

Thank you for the opportunity to resubmit our manuscript to *Nature Communications* for consideration for publication. For ease of reading we have directly pasted the Reviewer's comments below, which are in Calibri font. We also submitted a clean version of manuscript without tracked changes.

Our responses are in Times New Roman font, preceded by "**Response:**"

Reviewer #1 (Remarks to the Author):

I suggest an acceptance of the paper, but also want the authors to write the Data availability clear in the main text.

In the main text, the authors stated: "The raw sequence data reported in this paper have been deposited in the Genome Sequence Archive in National Genomics Data Center, China National Center for Bioinformation / Beijing Institute of Genomics, Chinese Academy of Sciences, under accession number HRA001385 that are publicly accessible at <https://ngdc.cncb.ac.cn/gsa-human>;" But in the response, the authors said: "We have tried to release the raw data freely, however, according to the regulations of the Human Genetic Resources Administration of China (HGRAC), our data should be controlled-access". I am very confused here. If the raw data are not publicly accessible, please make it clear in the main text to avoid any misunderstanding to the readers.

Response: Thanks so much that you suggest the manuscript is acceptable. And sorry for the misunderstanding, we have revised the Data Availability section (Page 48, Line 1052 - Line 1059). "The raw sequence data reported in this paper have been deposited in the Genome Sequence Archive in National Genomics Data Center, China National

Center for Bioinformation / Beijing Institute of Genomics, Chinese Academy of Sciences, under accession number HRA001385 (<https://ngdc.cncb.ac.cn/gsa-human/>). The allele frequency of all variants and genotype imputation server are freely available via the website (<https://wbbc.westlake.edu.cn>). The corresponding authors will provide additional help for accessing the data upon request. Other data are available within the manuscript, the Supplementary materials and the Source Data”

Reviewer #2 (Remarks to the Author):

Thank you for the hard work revising the manuscript. Most of the concerns in the previous revision have been addressed to my satisfaction.

As a minor note, with regards to our discussion of the use of the FREEMIX contamination estimate, the pipeline used here may be unfairly reducing the quality of the data. Supplying a constant value of 0.05 as the contamination value to HaplotypeCaller will lead to all variant calls effectively having 5% of their reads removed, which is roughly equivalent to reducing your sequencing coverage by 5%. The optimal way to use this value is to supply each HaplotypeCaller task with the value appropriate for each sample individually. I understand that making this change to the pipeline would require reanalyzing a large amount of data. There is nothing inaccurate about the method used here, but it's worth noting and pointing out that it's not quite in accordance with the GATK Best Practices.

Response: Thanks so much for the previous comments which help us a lot to improve the manuscript. We agreed that the constant value 0.05 would remove 0.2%-5% of reads. And your suggestion to apply appropriate value for each sample individually is reasonable. In fact, the WBBC project will continue to sequence more samples, and we will optimize the pipeline based on our previous practice and by taking the advices from experts like you, thanks a lot.

Reviewer #3 (Remarks to the Author):

The authors properly addressed most of my comments. However, there is one outstanding comment that still remains and must be address before the publication.

Original comment:

Line 57 : "and the selection of the alcohol metabolism genes (ADH1A and ADH1B) strengthened from about 4,000 years ago in East Asia". This claim does not have a credible basis. It was never described how these estimates were obtained throughout the manuscript.

Response:

We have added the analysis in the Method section (Page 48, Line 1037 - Line 1044). We estimated the selection coefficient trajectories for ADH1A (rs3819197), ADH1B (rs1229984) and ALDH2 (rs671) genes with a hidden Markov model using the ancient allele frequencies from East Asian ancient individuals (9,500 - 300 BP) and present-day allele frequencies in the WBBC. We divided ancient individuals into different generations (25 years per generation) according to their age and calculated derived allele frequency per generation. Allele frequency trajectories and selection coefficients for each gene were simulated by using a hidden Markov Model with an effective population size of 10,000. The derived allele of the SNP rs1229984 and rs3819197 was present around the 7,000 year ago, but was very rare for a long time (Fig. 2c). The strength of selection at the ADH1A (rs3819197) and ADH1B (rs1229984) gene increased in East Asian ancient individuals around 4,000 years ago.

New comment:

The description of the method is not detailed enough. The method is not published elsewhere, and the details of HMM is not clearly stated enough so I am unable to judge how credible the inference is. It is very hard to specify the exact year when the selection happened especially when the ancient genomes are not from the same region and when the ancient genomes are of limited sample size. Much more

detailed description of the algorithm is necessary to justify the conclusion, and I suggest to make a more simplistic conclusion or provide additional mathematical details (probably multiple pages) how the inference was made in a enough detail that the method and conclusion can be fairly evaluated and can be reproduced elsewhere.

The results that ADH1A and ADH1B was under strong selection is further questions based on the new results of SDS stratified by province. In that result, chr4 has barely any signals except for 1 other province. This suggest that population structure may played a role in the spurious selection signals. More sophisticated method is warranted to disentangle the selection effect from population structure between current and ancient genomes.

Response: Allele frequency time series data is a powerful resource for accounting for the strength of nature selection. Several studies have analyzed the nature selection based on the time series allele frequency data [1-4]. The method for estimating selection coefficients and allele frequency trajectory in our study was published in 2013 [5]. The authors of this method also used it to estimated allele frequency trajectories of *FADS1/FADS2* gene in the past 12,000 years in 2018 [6]. The derived allele of *FADS* gene was introduced ~8,500 years ago. They found that the selection in the *FADS* gene was not tightly linked to the initial introduction of agriculture and the Neolithic transition. The R package “mathii/salttice” based on this method is available on github (<https://github.com/mathii/slattice>). The workflow of this method was composed of three main steps: (1) choose model; (2) maximum-likelihood estimators; (3) estimation using hidden Markov models. In our study, we selected 204 ancient samples (9,500 - 300 BP) that covered SNP rs1229984 and 157 ancient samples that covered SNP rs3819197 from the 396 ancient individuals, we did not find ancient samples that covered SNP rs671. We also included 504 modern East Asian samples from 1000G project. We then divided ancient individuals into different generations (25 years per generation) according to their age and calculated derived allele

frequency per generation (for example: 9500-9475 years ago is the first generation contained 1 sample, 25 years ago - present is the 381 generation contained 504 modern samples). The allele frequency of each generation were then calculated as the input of the R package. We used the R packages “slattice” and defined the effective population size (N_e) as 10,000 and chose “Soft EM” Model in the estimation. We found that the derived alleles of *ADH1A* and *ADH1B* emerged around 7,000 year ago and tended to be more common from 4,000 years ago in the ancient population. Interestingly, Peng et al estimated the allele age in East Asian population and reported that the emergence of the *ADH1B* (rs1229984) occurred about 10,000~7,000 years ago [8], and another study suggested that the selection in the *ADH1B* gene intensified around 4,000 years ago in the northern East Asia [7]. We revised the Results, Discussion, and Method accordingly on Line 245, Line 551 and Line 943.

We agreed that the population structure might play a role in the selection signals and the imbalance distribution of samples is a limitation in our study. As we answered in the previous responses, in the 4334 Han Chinese individuals, 3074 individuals are from Hunan province, 703 are from Jiangxi province and 557 individuals are from other provinces. To address your comment here, we randomly extracted 350, 500, 600, 700, 800, 900, 1000, 1200, 1400, 1600, 1800 and 2000 individuals from Hunan province which have ~3000 samples. We observed that the selection signals were instable with insufficient samples, while the sign of selection could be observed as the number of samples increased (Figure 1 below). The significant selection signals on the *ADH1A* and *ADH1B* gene were constantly detected when the sample size was more than 1600 (Figure 1 below). As there are only 703 samples in Jiangxi province, we observed a trend of selection signal in this small sample size ($p=1.29E-05$ for *ADH1B*) (Figure 2 below). And we observed significant selection signal in other provinces ($p=2.05E-08$ for *ADH1B*) (Figure 3 below). In our study, besides the novel selection signals on *SNX29*, *DNAH1* and *WDR1*, the positive selection signatures on *ADH1A* and *ADH1B* were confirmed in the East Asian population [9] [10] and in the Japanese population [11]. No significant selection signal in the ADH-cluster gene was

detected in the population in Ulaanbaatar[12] and Tibetans[13]. We have made changes on Line 546 - Line 549 in Discussion.

Figure 1. Manhattan plot of the selection signatures for chromosome 4 of Hunan Provinces with different number of random selected individuals (350, 500, 600, 700, 800, 900, 1000, 1200, 1400, 1600, 1800 and 2000 individuals).

Figure 2. Manhattan plot of the selection signatures for 703 Jiangxi individuals.

Figure 3. Manhattan plot of the selection signatures for 557 individuals from other provinces.

Reference:

1. Bollback JP, York TL, Nielsen R. 2008. Estimation of $2N_e$ s from temporal allele frequency data. *Genetics*, 179:497-502.
2. Feder AF, Kryazhimskiy S, Plotkin JB. 2014. Identifying signatures of selection in genetic time series. *Genetics*, 196:509-522.
3. Ferrer-Admetlla A, Leuenberger C, Jensen JD, Wegmann D. 2016. An Approximate Markov Model for the Wright-Fisher Diffusion and Its Application to Time Series Data. *Genetics*, 203:831-846.
4. Malaspinas AS, Malaspinas O, Evans SN, Slatkin M. 2012. Estimating allele age and selection coefficient from time-serial data. *Genetics*, 192:599-607.
5. Mathieson I, McVean G. 2013. Estimating selection coefficients in spatially structured populations from time series data of allele frequencies. *Genetics*, 193:973-984.
6. Mathieson S, Mathieson I. 2018. FADS1 and the Timing of Human Adaptation to Agriculture. *Mol Biol Evol*, 35:2957-2970.
7. Mathieson I. 2020. Estimating time-varying selection coefficients from time series data of allele frequencies. *bioRxiv:2020.2011.2017.387761*.
8. Peng Y, Shi H, Qi XB, Xiao CJ, Zhong H, Ma RL, Su B. 2010. The ADH1B Arg47His polymorphism in east Asian populations and expansion of rice domestication in history. *BMC Evol Biol*, 10:15.
9. Voight BF, Kudaravalli S, Wen X, Pritchard JK. 2006. A map of recent positive selection in the human genome. *PLoS Biol*, 4:e72.
10. Han Y, Gu S, Oota H, Osier MV, Pakstis AJ, Speed WC, Kidd JR, Kidd KK. 2007. Evidence of positive selection on a class I ADH locus. *Am J Hum Genet*, 80:441-456.
11. Okada Y, Momozawa Y, Sakaue S, Kanai M, Ishigaki K, Akiyama M, Kishikawa T, Arai Y, Sasaki T, Kosaki K, Suematsu M, Matsuda K, Yamamoto K, Kubo M, Hirose N, Kamatani Y. 2018. Deep whole-genome sequencing reveals recent selection signatures linked to evolution and disease risk of Japanese. *Nat Commun*, 9:1631.

12. Nakayama K, Ohashi J, Watanabe K, Munkhtulga L, Iwamoto S. 2017. Evidence for Very Recent Positive Selection in Mongolians. *Mol Biol Evol*, 34:1936-1946.
13. Lu Y, Kang L, Hu K, Wang C, Sun X, Chen F, Kidd JR, Kidd KK, Li H. 2012. High diversity and no significant selection signal of human ADH1B gene in Tibet. *Investig Genet*, 3:23.

Reviewers' Comments:

Reviewer #3:

Remarks to the Author:

The authors adequately addressed my remaining comments, and I am happy to recommend accepting the revised manuscript.

Title: Genomic analyses of 10,376 individuals in the Westlake BioBank for Chinese (WBBC) pilot project

Our responses are in Times New Roman font, preceded by “**Response:**”

Reviewer #3 (Remarks to the Author):

The authors adequately addressed my remaining comments, and I am happy to recommend accepting the revised manuscript.

Response: Thanks so much for the all your comments which help us a lot to improve the manuscript.